# Stability and Generalization of Adversarial Training for Shallow Neural Networks with Smooth Activation

**Kaibo Zhang**
Johns Hopkins University
Baltimore, MD 21218
kzhang90@jhu.edu

**Yunjuan Wang**
Johns Hopkins University
Baltimore, MD 21218
ywang509@jhu.edu

**Raman Arora**
Johns Hopkins University
Baltimore, MD 21218
arora@cs.jhu.edu

## Abstract

Adversarial training has emerged as a popular approach for training models that are robust to inference-time adversarial attacks. However, our theoretical understanding of why and when it works remains limited. Prior work has offered generalization analysis of adversarial training, but they are either restricted to the Neural Tangent Kernel (NTK) regime or they make restrictive assumptions about data such as (noisy) linear separability or robust realizability. In this work, we study the stability and generalization of adversarial training for two-layer networks **without any data distribution assumptions** and **beyond the NTK regime**. Our findings suggest that for networks with *any given initialization* and *sufficiently large width*, the generalization bound can be effectively controlled via early stopping. We further improve the generalization bound by leveraging smoothing using Moreau's envelope.

## 1 Introduction

Despite the remarkable performance of over-parameterized deep networks in real-world applications, recent studies have revealed that they are highly vulnerable to adversarial attacks. These attacks use maliciously crafted imperceptible perturbations designed to deceive trained neural networks during inference [Szegedy et al., 2013, Biggio et al., 2013]. The lack of adversarial robustness has raised significant concerns for deploying neural network-based models in safety-critical applications. Therefore, it is crucial to design algorithms to learn robust models that can make reliable predictions on test data even in the presence of adversarial perturbations.

One principal approach to robust learning, adversarial training [Madry et al., 2018] (along with its variants [Zhang et al., 2019, Wang et al., 2020]), has proven to be an effective empirical defense mechanism against adversarial attacks. Naturally, this puts an emphasis on also developing a theoretical understanding of robust learning. To study the generalization performance of robust learning, one traditional approach is via uniform convergence [Khim and Loh, 2018, Yin et al., 2019, Awasthi et al., 2020, Mustafa et al., 2022], which provides the worst-case type uniform bounds for a given hypothesis class and are algorithm independent. Another line of work focuses on analyzing the convergence and generalization guarantees of adversarial training, yet they either focus on linear classifiers [Charles et al., 2019, Li et al., 2020, Zou et al., 2021, Chen et al., 2023], or introduce restrictive distribution assumptions such as (noisy) linear separability [Wang et al., 2024b] or robust realizability [Mianjy and Arora, 2024]. Therefore, it remains unclear whether we can derive theoretical results for adversarial training that extend beyond these simplifying assumptions.

In this work, we leverage a different machinery by analyzing adversarial training algorithm through the lens of uniform stability. Stability is a classical tool in learning theory that has been extensively studied in the literature [Bousquet and Elisseeff, 2002, Hardt et al., 2016]. Uniform argument stability measures the difference in output parameters when an algorithm is run on two training sets that

differ by only one sample. In the standard (non-robust) setting, Hardt et al. [2016] show a uniform stability bound of $\mathcal{O}(\frac{\eta T}{n})$ after $T$ iterations of gradient descent with step size $\eta$ on convex and smooth losses using a training dataset of size $n$. They further provide a uniform stability bound of $\mathcal{O}(\frac{T^q}{n})$ for smooth and non-convex losses with decaying step size $\eta = \mathcal{O}(\frac{1}{t})$, where $q \in (0, 1)$ is a constant. The choice of decaying step size is common in the non-convex setting, as maintaining a constant step size leads to an exponentially increasing bound on uniform stability.

When it comes to the robust setting, the primary challenge lies in the non-smoothness of the robust (adversarial) loss. The robust loss is generally non-smooth even if the standard counterpart is smooth [Xing et al., 2021a, Xiao et al., 2022a]. Previous work by Xing et al. [2021a] studied the convex non-smooth adversarial losses and provide an additional term of $\mathcal{O}(\eta\sqrt{T})$ compared to the convex and smooth losses. Later Xiao et al. [2022a] studied the general non-smooth adversarial losses by leveraging the approximate co-coercivity of the gradient and provide the bound with an additional term of $\mathcal{O}(\eta T \alpha)$ that grows linearly in $T$, where $\alpha$ is the size of adversarial perturbation in $\ell_p$ threat models. These works, while partially addressing the issue, only focus on general convex / non-convex functions. However, neural networks, which are a specific instance of non-convex functions and are widely used in practice, require further investigation.

In this work, we study the stability and generalization guarantees of variants of adversarial training algorithms. We focus on solving the binary classification problem using two-layer over-parameterized neural networks with smooth activation functions and logistic loss. Our key contributions are as follows:

1. We present a bound of $\mathcal{O}(\sqrt{\eta}T + \frac{\eta T}{n} + \sqrt{\beta\eta T})$ on the uniform argument stability of the gradient descent-based adversarial training of over-parameterized network after $T$ iterations with step size $\eta$, where $\beta$ represents the precision of generating adversarial examples at each iteration.

2. We provide robust generalization guarantees that depend on the Adversarial Regularized Empirical Risk Minimization (ARERM) Oracle. Our results hold for any given initialization and any data distribution. Specifically, if the learner is provided with a good initialization such that there exist robust networks around this initialization, then a small robust test loss is achieved via early stopping. Furthermore, our results can be extended to stochastic gradient descent-based adversarial training.

3. We leverage Moreau's envelope to construct a smooth loss that approximates robust empirical loss. We present bounds on the stability and generalization error of gradient descent with Moreau's smoothing, and demonstrate its superiority compared with gradient descent-based adversarial training algorithm.

### 1.1 Related Work

**Stability Analysis.** The notion of stability was initially introduced in Bousquet and Elisseeff [2002] to study the generalization of statistical learning problems. More recently, a fine-grained analysis has been presented by Feldman and Vondrak [2019] and Bousquet et al. [2020]. For smooth loss functions, Hardt et al. [2016] explored the stability of SGD in both convex and non-convex settings, which was later extended to convex non-smooth loss functions by Bassily et al. [2020] and the bound incorporated an additional term of $\mathcal{O}(\eta\sqrt{T})$ due to non-smoothness. Lei and Ying [2020] tackled the non-smoothness differently by assuming the gradient of the loss to be Hölder continuous. For non-convex and non-smooth loss, Lei [2023] introduced the stability of sub-gradient, as convergence to local minimizers is observed in this setting.

**Robust Generalization Guarantee.** The standard method of giving a generalization guarantee is through uniform convergence. These theories typically yield an upper bound of $\mathcal{O}(\frac{1}{\sqrt{n}})$ and require a large number of training samples in order to get a small generalization gap. Techniques in this category include analyzing the Rademacher complexity [Yin et al., 2019, Khim and Loh, 2018, Awasthi et al., 2020], VC dimension [Cullina et al., 2018, Montasser et al., 2020], covering number [Balda et al., 2019, Mustafa et al., 2022, Li and Telgarsky, 2023], PAC Bayesian analysis [Farnia et al., 2018, Viallard et al., 2021, Xiao et al., 2023] and margin-based analysis [Farnia et al., 2018].

**Generalization Guarantee of Adversarial Training.** Providing generalization guarantees for adversarial training of neural networks is challenging due to its non-convex nature. A series of

works [Charles et al., 2019, Li et al., 2020, Zou et al., 2021, Chen et al., 2023] have focused on a simpler problem – adversarial training of linear models with a convex loss wherein generating adversarial examples admits a closed-form solution. Several works bypass this challenge by considering a lazy training regime [Gao et al., 2019, Zhang et al., 2020, Li and Telgarsky, 2023] in which the landscape of the neural network can be studied near certain random initialization, and the generalization guarantee is usually obtained via uniform convergence. Unfortunately, Wang et al. [2022] proved that adversarial robustness is at odds with lazy regime. Recently, Mianjy and Arora [2024], Wang et al. [2024b] provide convergence and generalization guarantees for adversarial training of neural networks, yet they make restrictive assumptions on the data distribution such as (noisy) linear separability and robust realizability.

Another line of research investigates the generalization of adversarial training through algorithmic stability analysis. Despite the smoothness of the standard loss, the adversarial loss remains non-smooth [Liu et al., 2020, Xing et al., 2021a, Xiao et al., 2022b]. To resolve this issue, Farnia and Ozdaglar [2021] make a strong assumption that the loss is concave in input x. Xing et al. [2021a] provide adversarial training of convex and non-smooth losses, yielding an additional term of $\mathcal{O}(\sqrt{\eta^2 T})$ compared to the standard non-robust counterpart. Xiao et al. [2022a] and Wang et al. [2024a] leverage the idea of approximate smoothness and provide bounds that scale linearly with $\eta T$ and the perturbation size. Cheng et al. [2024] consider generating adversarial examples via a single step of gradient descent and demonstrate that such variant of adversarial training algorithm achieves better stability. Farnia et al. [2018] also consider specific attack algorithms – these attacks while being more practical and designed with continuity and Lipschitzness property, may differ significantly from the worst-case attack, and do not yield a good bound on the robust generalization gap.

## 2  Problem Setup

**Notation.**  Throughout the paper, we denote scalars, vectors, and matrices with lowercase italics, lowercase bold, and uppercase bold Roman letters, respectively; e.g., $u$, u, and U. We use $[m]$ to denote the set $\{1, 2, \ldots, m\}$ and use both $\|\cdot\|$ and $\|\cdot\|_2$ for $\ell_2$-norm. Given a matrix $U = [u_1, \ldots, u_m] \in \mathbb{R}^{d \times m}$, we use $\|U\|_F$ and $\|U\|_2$ to represent the Frobenius norm and spectral norm, respectively. We use the standard O-notation ($\mathcal{O}$, $\Theta$ and $\Omega$).

We consider a binary classification problem with a bounded input space $\mathcal{X}$ inside a Euclidean ball of radius $C_x$, and label space $\mathcal{Y} = \{\pm 1\}$. We assume that data are drawn according to an unknown probability distribution $\mathcal{D}$ on $\mathcal{X} \times \mathcal{Y}$. The learner has access to $n$ training data drawn i.i.d. from $\mathcal{D}$; i.e., $S = \{z_i = (x_i, y_i)\}_{i=1}^n \sim \mathcal{D}^n$. We do not make any restrictive distributional assumptions such as realizability [Mianjy and Arora, 2024] or (noisy) linearly separability [Wang et al., 2024b].

We focus on learning two-layer neural networks, parameterized by a pair of weight matrices $(a, W)$:
$$f_W(x) = f(x; W) := \sum_{s=1}^m a_s \phi(\langle w_s, x \rangle).$$
Here, $m$ is a positive integer representing the number of hidden units, i.e., the width of the networks. $\phi : \mathbb{R} \to \mathbb{R}$ is a 1-Lipschitz, $H$-smooth activation function. Formally, $\forall z, z' \in \mathbb{R}, |\phi'(z)| \leq 1, |\phi'(z) - \phi'(z')| \leq H|z - z'|$. The smoothness property of activation functions is commonly assumed in algorithmic stability literature and in theory of deep learning and covers a wide range of activation functions such as smoothed ReLU and smoothed leaky ReLU [Frei et al., 2022]. The weight matrices at the top and bottom layer are denoted as $a = [a_1, \ldots, a_m] \in \mathbb{R}^m$ and $W = [w_1, \ldots, w_m] \in \mathbb{R}^{d \times m}$, respectively. The top layer weights are initialized such that $|a_i| = \frac{1}{\sqrt{m}}, \forall i \in [m]$, and are kept fixed throughout the training process. Prior works [Du et al., 2018, Arora et al., 2019, Ji and Telgarsky, 2019] often initialize $a_i$ to be uniformly sampled from $\{\pm \frac{1}{\sqrt{m}}\}$, which can be seen as a special instance of ours. We do not make any assumption on the initialization of the bottom layer matrix, i.e., $W_0$ can be either a standard Gaussian [Du et al., 2018, Ji and Telgarsky, 2019], or a vanishing initialization [Ba et al., 2019, Xing et al., 2021b], or a pre-trained model.

**Adversarial Attacks.**  We consider a general threat model where the adversary's perturbation set is defined as $\mathcal{B} : \mathcal{X} \to 2^{\mathcal{X}}$. Given an input x, $\mathcal{B}(x)$ represents the set of all possible perturbations of x that an adversary can choose from. This broader definition of attack includes both the standard $\ell_p$ threat models with perturbation size of $\alpha$, i.e., $\mathcal{B}(x) = \{\tilde{x} : \|\tilde{x} - x\|_p \leq \alpha\}$, as well as a discrete set of large-norm transformations. Unlike prior works [Mianjy and Arora, 2024, Wang et al., 2024b], we do not make any assumptions on the perturbation size.

In this work, we focus on logistic loss, $\ell(z) = \ln(1 + e^{-z})$, which serves as a smooth and convex surrogate loss for the 0-1 loss. With a slight abuse of notation, for a fixed sample $z = (x, y)$, we define $\ell(z, W) := \ell(yf(x; W))$. The population and empirical loss w.r.t. $\ell(\cdot)$ are denoted, respectively, as

$$L(W) := \mathbb{E}_{(x,y)\sim\mathcal{D}}\ell(yf(x; W)), \quad \widehat{L}(W; S) := \frac{1}{n}\sum_{i=1}^{n}\ell(y_i f(x_i; W)).$$

Given $\mathcal{B}$, for a fixed sample $z = (x, y)$, we define the robust loss as $\ell_{rob}(z, W) := \max_{\tilde{x}\in\mathcal{B}(x)}\ell(yf(\tilde{x}; W))$.

The robust population and empirical loss w.r.t. $\ell(\cdot)$ are defined as

$$L_{rob}(W) := \mathbb{E}_{(x,y)\sim\mathcal{D}}\max_{\tilde{x}\in\mathcal{B}(x)}\ell(yf(\tilde{x}; W)) \quad \widehat{L}_{rob}(W; S) := \frac{1}{n}\sum_{i=1}^{n}\max_{\tilde{x}_i\in\mathcal{B}(x_i)}\ell(y_i f(\tilde{x}_i; W)).$$

**Adversarial Training.** During training, the network bottom layer weight W are updated using gradient descent-based adversarial training (or its stochastic version). We denote the weight matrix at the $t$-th iterate of adversarial training as $W_t$. For each training example $(x_i, y_i)$, at iteration $t$, we generate a $\beta_1$-*optimal adversarial example* $(\tilde{x}_i(W_t), y_i)$, which satisfies the following condition:

$$\ell(y_i f(\tilde{x}_i(W_t); W_t)) \geq \max_{\tilde{x}\in\mathcal{B}(x_i)}\ell(y_i f(\tilde{x}; W_t)) - \beta_1. \tag{1}$$

Setting $\beta_1 = 0$ recovers the scenario where we have access to the worst-case adversarial attack. As this may not be feasible in practice due to computational reason, the parameter $\beta_1$ allows us to capture the precision of the attack algorithm, which includes common attacks such as projected gradient descent (PGD) [Madry et al., 2018]. We should regard $\beta_1$ as a parameter we can choose. Our results in Section 3 suggest that we can achieve better generalization by adding more computation and making $\beta_1$ smaller.

---

**Algorithm 1** Variants of Adversarial Training Algorithms

---

**Input:** Step size $\eta$. Number of iterations $T$. Initial weight $W_0$. $\beta \geq 0$. $\mu > 0$.
    **for** $t = 0, \ldots, T-1$ **do**
        **GD:** $\forall i \in [n]$, compute a $\beta_1$-optimal adversarial example $\tilde{x}_i(W_t)$ that satisfies Equation (1).
            Update $W_{t+1} = W_t - \frac{\eta}{n}\sum_{i=1}^{n}\nabla_W\ell(y_i f(\tilde{x}_i(W_t); W_t))$.
        **SGD:** Compute a $\beta_1$-optimal adversarial example $\tilde{x}_{t+1}(W_t)$ that satisfies Equation (1).
            Update $W_{t+1} = W_t - \eta\nabla_W\ell(y_{t+1}f(\tilde{x}_{t+1}(W_t); W_t))$.
        **Moreau Envelope:** Compute a $\beta_2$-optimal minimizer $\widetilde{U}^{\mu}(W_t; S)$ that satisfies Equation (2).
            Update $W_{t+1} = W_t - \frac{\eta}{\mu}(W_t - \widetilde{U}^{\mu}(W_t; S))$.
    **end for**
    return: $\{W_t\}_{t=0}^{T}$.

---

**Optimizing the Moreau Envelope.** Since the robust loss is non-smooth [Xiao et al., 2022a], we utilize Moreau's envelope to construct a smooth function that approximates the empirical robust loss. Such an idea has previously been explored in Xiao et al. [2024]. Given training data $S$ and $\mu > 0$, we redefine the robust surrogate loss as follows:

$$M^{\mu}(W; S) = \min_U\left(\widehat{L}_{rob}(U; S) + \frac{1}{2\mu}\|U - W\|_F^2\right).$$

Selecting $\mu$ appropriately ensures that $\widehat{L}_{rob}(U; S) + \frac{1}{2\mu}\|U - W\|_F^2$ is a strongly convex function w.r.t. U. Given W and $S$, we define $U^{\mu}(W; S) = \operatorname{argmin}_{U\in\mathbb{R}^{d\times m}}\widehat{L}_{rob}(U; S) + \frac{1}{2\mu}\|U - W\|_F^2$, which can be obtained via subgradient-based method (solve a min-max optimization). The gradient of the Moreau envelope can be simply calculated as $\nabla_W M^{\mu}(W; S) = \frac{1}{\mu}(W - U^{\mu}(W; S))$. Given training data $S$, at each iteration $t$, we generate a $\beta_2$-*optimal minimizer* $\widetilde{U}^{\mu}(W_t; S)$ that satisfies

$$\widehat{L}_{rob}(\widetilde{U}^{\mu}(W_t; S); S) + \frac{1}{2\mu}\|\widetilde{U}^{\mu}(W_t; S) - W_t\|_F^2 \leq \beta_2 + M^{\mu}(W_t; S). \tag{2}$$

We remark that $\beta_2$-optimal minimizer defined in Equation (2) and $\beta_1$-optimal adversarial example defined in Equation (1) are approximating different quantities, which are not comparable. All the algorithms described above are summarized in Algorithm 1.

**Uniform Argument Stability.** Given a training set $S = \{z_i\}_{i=1}^n$ drawn i.i.d. from $\mathcal{D}$, let $S'$ denote the training set obtained by replacing one example in $S$ with an independently drawn example $z' \sim \mathcal{D}$. We refer to $S, S'$ as neighboring samples and write $S \simeq S'$. Given an algorithm $\mathcal{A} : (\mathcal{X} \times \mathcal{Y})^n \to \mathcal{H}$, where the hypothesis class $\mathcal{H}$ is parameterized using a parameter matrix $W \in \mathbb{R}^{d \times m}$, we define the uniform argument stability as

$$\delta_{\mathcal{A}}(S, S') := \|\mathcal{A}(S) - \mathcal{A}(S')\|_F.$$

For any $L$-Lipschitz loss function $g$, $|g(\mathcal{A}(S), z) - g(\mathcal{A}(S'), z)| \le L\delta_{\mathcal{A}}(S, S')$. The standard stability argument [Mohri et al., 2018] relates the expected generalization gap to the uniform argument stability.

$$\mathbb{E}_{S \sim \mathcal{D}^n} \varepsilon_{gen}(\mathcal{A}(S)) := \mathbb{E}_{S \sim \mathcal{D}^n} \left( \mathbb{E}_{z \sim \mathcal{D}} g(\mathcal{A}(S), z) - \frac{1}{n} \sum_{i=1}^n g(\mathcal{A}(S), z_i) \right) \le L \sup_{S \simeq S'} \delta_{\mathcal{A}}(S, S'). \quad (3)$$

In this paper, we consider robust generalization using logistic loss, so function $g(W, z) = \ell_{rob}(z, W)$, and $\varepsilon_{gen}(W) = \mathbb{E}_{z \sim \mathcal{D}}[\ell_{rob}(z, W)] - \frac{1}{n} \sum_{i=1}^n \ell_{rob}(z_i, W)$. We also remark that a high probability bound for stable algorithms can be given based on Feldman and Vondrak [2019]. For simplicity, our generalization bounds in this paper are only in expectation.

# 3 Main Result

In this section, we present our main results, providing theoretical guarantees for adversarial training of two-layer neural networks with smooth activation functions. We discuss (stochastic) adversarial training in Section 3.1 and gradient descent-based Moreau's smoothing in Section 3.2. Our generalization bounds rely on a key quantity, the *Adversarial Regularized Empirical Risk Minimization (ARERM) Oracle* defined as

$$\Delta_S^{\text{oracle}} := \min_{W \in \mathbb{R}^{d \times m}} \left( \widehat{L}_{rob}(W; S) + \frac{2\|W - W_0\|_F^2}{\eta T} \right).$$

Given a sample, $\Delta_S^{\text{oracle}}$ returns the minimal empirical risk in the vicinity of an initialization $W_0$.

## 3.1 Generalization Guarantees for Adversarial Training

We begin by presenting a bound on the uniform argument stability (UAS) of Algorithm 1 with GD.

**Theorem 3.1.** Assume that the network width satisfies $m \ge H^2 C_x^4 \eta^2 (T + 1)^2$. Then, after $T$ iterations of Algorithm 1 with GD, for any neighboring datasets $S, S'$, we have

$$\sup_{S \simeq S'} \delta_{\mathcal{A}}(S, S') \le \mathcal{O}(C_x \eta \sqrt{T} + C_x \frac{\eta T}{n} + \sqrt{\beta_1 \eta T}).$$

Remarkably, setting $\beta_1 = 0$ yields a bound of $\mathcal{O}(\eta\sqrt{T} + \frac{\eta T}{n})$ on the UAS of Algorithm 1, thereby recovering the result in prior work of Xing et al. [2021a]. However, note that Xing et al. [2021a] show the result only for convex learning problems, whereas we consider training two-layer neural networks using logistic loss, which is non-convex and non-smooth. Further note that we assume that the networks are sufficiently over-parameterized, i.e., $m \ge \Omega(\eta^2 T^2)$, a condition that is commonly assumed in deep learning theory. We can also regard this condition as early stopping, wherein $T \le \mathcal{O}\left(\frac{\sqrt{m}}{HC_x^2 \eta}\right)$. This view is also consistent with several empirical studies [Caruana et al., 2000, Rice et al., 2020, Pang et al., 2021].

Next, we show that stable robust learning rules do not overfit.

**Theorem 3.2.** Define $\alpha_1(\eta, T) := \mathcal{O}(C_x^2 \eta \sqrt{T} + C_x^2 \frac{\eta T}{n} + C_x \sqrt{\beta_1 \eta T})$. Assume that the width of the networks satisfies $m \ge H^2 C_x^4 \eta^2 (T + 1)^2$, and $\alpha_1(\eta, T) < 1$. Then, after $T$ iterations of Algorithm 1 with GD, we have

$$\min_{\lceil \frac{9T}{10} \rceil \le t \le T} \mathbb{E}_{S \sim \mathcal{D}^n} \varepsilon_{gen}(W_t) \le \frac{17\alpha_1(\eta, T)}{1 - \alpha_1(\eta, T)} \left[ \mathbb{E}_{S \sim \mathcal{D}^n} \Delta_S^{\text{oracle}} + \frac{C_x^2 \eta}{2} + \beta_1 \right],$$

and

$$\min_{0 \le t \le T} \mathbb{E}_{S \sim \mathcal{D}^n} L_{rob}(W_t) \le \frac{1}{1 - \alpha_1(\eta, T)} \left[ \mathbb{E}_{S \sim \mathcal{D}^n} \Delta_S^{\text{oracle}} + \frac{C_x^2 \eta}{2} + \beta_1 \right].$$

The result above bounds the robust generalization gap and the robust loss in terms of the ARERM oracle, a step size-dependent term $\mathcal{O}(\eta)$, and the precision of the adversarial examples $\beta_1$. Note though that the bound holds for the minimum over the last few iterates (past iterates), rather than for the last iteration. This distinction arises because, unlike standard gradient descent for neural networks, we cannot guarantee a decreasing robust training loss without additional assumptions on the data distributions owing to the non-smooth nature of the robust loss. The step size-dependent term arises for the same reason. A direct corollary gives us a bound on the expected robust loss.

**Corollary 3.3.** After $T \leq \mathcal{O}(\min\{n^2, \frac{1}{\beta_1^2}\})$ iterations of Algorithm 1 with GD using a step size of $\eta = \Theta(\frac{1}{C_x^2\sqrt{T}})$ on a network with width $m \geq \Omega(T)$, for any weight matrix W

$$\min_{0 \leq t \leq T} \mathbb{E}_{S \sim \mathcal{D}^n} L_{rob}(\mathbf{W}_t) \leq 1.1 L_{rob}(\mathbf{W}) + \mathcal{O}\left(\frac{C_x^2 \|\mathbf{W} - \mathbf{W}_0\|_F^2}{\sqrt{T}}\right) + \mathcal{O}\left(\frac{1}{\sqrt{T}}\right).$$

Since corollary 3.3 holds for any $\mathbf{W}_0$, it underscores the importance of initialization for robust learning. Given a good initialization, such as a pre-trained model, and assuming that there exists a robust network $\mathbf{W}_*$ in the vicinity of the initialization (i.e., $\|\mathbf{W}_* - \mathbf{W}_0\|_F = \mathcal{O}(1)$) that achieves a small robust loss $L_{rob}(\mathbf{W}_*) \approx 0$, we have that the minimum expected robust loss over all iterates approaches $\mathcal{O}\left(\frac{1}{\sqrt{T}}\right)$. Further, if $\beta_1$ is small enough and $m \gtrsim n^2$, then $T$ can be of the order $\Theta(n^2)$, leading to a $\mathcal{O}(1/n)$ upper bound on the robust test loss.

We remark that by a similar analysis, our result can be reduced to the standard (non-robust) setting for gradient descent training of two-layer networks by setting the perturbation set $\mathcal{B}(\mathbf{x}) = \{\mathbf{x}\}, \forall \mathbf{x} \in \mathcal{X}$, $\beta_1 = 0$, and redefining $\alpha_1(\eta, T) = \mathcal{O}(C_x^2 \frac{\eta T}{n})$. In this context, we can show that gradient descent for the binary classification problem can achieve excess risk bound of $\mathcal{O}(1/\sqrt{n})$ by taking $\eta T = \Theta(\sqrt{n})$ if $m \gtrsim n$ and assuming $\|\mathbf{W}_* - \mathbf{W}_0\|_F = \mathcal{O}(1)$, where $\mathbf{W}_* \in \operatorname{argmin}_{\mathbf{W}} L_{rob}(\mathbf{W})$.

Next, we extend our result to the stochastic adversarial training.

**Theorem 3.4.** After $T$ iterations of Algorithm 1 with SGD on a network of width $m \geq H^2 C_x^4 \eta^2 (T + 1)^2$ we have that for any weight matrix W,

$$\min_{0 \leq t \leq T} \mathbb{E}_{\{z_1, \ldots, z_t\} \sim \mathcal{D}^t} L_{rob}(\mathbf{W}_t) \leq L_{rob}(\mathbf{W}) + \frac{\|\mathbf{W} - \mathbf{W}_0\|_F^2}{\eta(T + 1)} + \frac{C_x^2 \eta}{2} + \beta_1.$$

Similar to the discussion following Corollary 3.3, we assert that if we assume that there exists an over-parameterized robust network with small robust loss, then using a step size of $\eta = 1/\sqrt{T}$, stochastic adversarial training yields an excess risk bound of $\mathcal{O}(1/\sqrt{T})$.

## 3.2 Generalization Guarantees for Gradient Descent on Moreau's Envelope

We now present a bound on the uniform argument stability of gradient descent with smoothing based on Moreau's envelope.

**Theorem 3.5.** After $T$ iterations of Algorithm 1 with Moreau Envelope with step-size $\eta \leq \min\{\mu, \frac{\sqrt{m}}{8HC_x^2}\} \leq \frac{\sqrt{m}}{2HC_x^2}$, on a network of width $m \geq H^2 C_x^4 \eta^2 T^2$, for any neighboring datasets $S, S'$, we have

$$\sup_{S \simeq S'} \delta_{\mathcal{A}}(S, S') \leq \mathcal{O}\left(C_x \frac{\eta T}{n} + \eta T \sqrt{\frac{\beta_2}{\mu}}\right).$$

Setting $\beta_2 = 0$ yields a bound of $\mathcal{O}(\frac{\eta T}{n})$ on the UAS of Algorithm 1, thereby recovering the result in prior work of [Hardt et al., 2016, Xiao et al., 2024] for convex and smooth functions. Note that by using Moreau's envolope, we are able to shave off the $\mathcal{O}(\eta\sqrt{T})$ term that appears in Theorem 3.1.

Although inspired by Xiao et al. [2024], Theorem 3.5 differs from the non-convex setting of Xiao et al. [2024]. Our result utilizes the specific structure of over-parameterized neural networks that exhibit weakly convex properties, a special instance of non-convex functions, and allows for a constant step size. In contrast, [Xiao et al., 2024, Theorem 4.7] follows the traditional stability argument for non-convex and smooth functions in Hardt et al. [2016], considering a decaying step size $\eta_t \leq \frac{\mu}{t}$.

Such a condition might be impractical if $\mu$ is chosen to be sufficiently small. In fact, our results indicate that it is necessary to select a sufficiently small $\mu$ so that the robust training loss is well approximated by the Moreau envelope (see Lemma C.1 in the Appendix).

Even though the gradient descent-based algorithm with Moreau's smoothing achieves better stability guarantees compared to gradient descent-based adversarial training when $\beta_1 = \beta_2 = 0$, it requires more computational resources. Specifically, for the calculation of the gradient at each step, we need to solve a min-max optimization problem with a strongly convex and non-smooth objective to obtain a $\beta$-optimal minimizer. Additionally, for every step of this min-max optimization, we need to generate adversarial examples and apply sub-gradient descent.

**Theorem 3.6.** Define $\alpha_2(\eta, T) := \mathcal{O}(C_x^2 \frac{\eta T}{n} + C_x \eta T \sqrt{\frac{\beta_2}{\mu}})$. Assume $\alpha_2(\eta, T) < 1$. Then, after $T \geq 8$ iterations of Algorithm Moreau Envelope with step-size $\eta \leq \mu$ on a network of width $m \geq H^2 C_x^4 \eta^2 T^2$, we have

$$
\min_{[\frac{9T}{10}] \leq t \leq T} \mathbb{E}_{S \sim \mathcal{D}^n} \varepsilon_{gen}(\mathbf{W}_t) \leq \frac{55 \alpha_2(\eta, T)}{1 - \alpha_2(\eta, T)} \left[ \mathbb{E}_{S \sim \mathcal{D}^n} \Delta_S^{\text{oracle}} + C_x^2 \mu + 2\eta(T+1)\frac{\beta_2}{\mu} \right],
$$

and

$$
\min_{1 \leq t \leq T} \mathbb{E}_{S \sim \mathcal{D}^n} L_{rob}(\mathbf{W}_t) \leq \frac{1}{1 - \alpha_2(\eta, T)} \left[ \mathbb{E}_{S \sim \mathcal{D}^n} \Delta_S^{\text{oracle}} + C_x^2 \mu + 2\eta(T+1)\frac{\beta_2}{\mu} \right].
$$

Similar to Theorem 3.2, the result above shows that both the robust generalization gap as well as the robust loss can be bounded in terms of the ARERM oracle, parameter $\mu$ in Moreau's envelope, and a term of $\mathcal{O}(\eta T \beta_2/\mu)$ dependent on the precision of generating the minimizer of Moreau envelope. While the bound above is on the minimum expected generalization gap (and expected robust test loss) over the last few iterates (past iterates), we can give a bound for the the last iterate for the case when $\beta_2 = 0$. We conclude the section by presenting the following direct corollary.

**Corollary 3.7.** After $T \leq \mathcal{O}(\min\{n^2, \frac{1}{\beta_2^{2/3}}\})$ iterations of Algorithm 1 with Moreau Envelope with step-size $\eta = \mu = \Theta(\frac{1}{C_x^2 \sqrt{T}})$ on a network of width $m \geq \Omega(T)$, we have for any weight matrix $\mathbf{W}$,

$$
\min_{1 \leq t \leq T} \mathbb{E}_{S \sim \mathcal{D}^n} L_{rob}(\mathbf{W}_t) \leq 1.1 L_{rob}(\mathbf{W}) + \mathcal{O}\left( \frac{C_x^2 \|\mathbf{W} - \mathbf{W}_0\|_F^2}{\sqrt{T}} \right) + \mathcal{O}\left( \frac{1}{\sqrt{T}} \right).
$$

## 4  Proof Sketch

We begin by providing a high level intuition behind our analysis technique, and then we highlight the key ideas in the proofs of the main theorems. For simplicity, we assume that the learner can generate optimal attacks during adversarial training, i.e., we consider $\beta_1 = 0$, $\beta_2 = 0$ in this section. We refer the reader to the Appendix for proofs of the more general case.

Our analysis relies on a key lemma demonstrating that the objective function (i.e., the robust empirical risk) being minimized in adversarial training of two-layer neural networks with smooth activation functions using the logistic loss function is "almost" convex.

**Definition 4.1.** Let $l > 0$. A function $f(x)$ is said to be $-l$-weakly convex if $f(x) + \frac{l}{2}\|x\|_2^2$ is convex in $x$.

**Lemma 4.2.** (Restatement of Lemma A.4) For any weight matrices $\mathbf{W}^1$ and $\mathbf{W}^2$,

$$
\widehat{L}_{rob}(\mathbf{W}^2; S) \geq \widehat{L}_{rob}(\mathbf{W}^1; S) + \left\langle \nabla_{\mathbf{W}} \widehat{L}_{rob}(\mathbf{W}^1; S), \mathbf{W}^2 - \mathbf{W}^1 \right\rangle - \frac{H C_x^2}{2\sqrt{m}} \|\mathbf{W}^2 - \mathbf{W}^1\|_F^2.
$$

Equivalently, $\widehat{L}_{rob}(\mathbf{W}; S)$ is $-\frac{H C_x^2}{\sqrt{m}}$-weakly convex.

We borrow many ideas from Xiao et al. [2024] and Xing et al. [2021a] in our proofs. These papers primarily focus on the convex setting, while only giving a general result for non-convex functions. We extend their results to a special case of learning neural networks. We argue that by specializing our analysis to neural networks would lead to sharper results than a general non-convex function class, as we will be able to leverage the "almost" convexity of neural network training [Richards

and Rabbat, 2021, Richards and Kuzborskij, 2021]. This allows us to get stability and optimization guarantees that are similar to the convex setting when we consider an over-parameterized network $m \geq \text{poly}(\eta T)$. An additional challenge we face is that the robust loss is non-smooth even if its standard counterpart (logistic loss) is smooth, making the analysis more complicated than the standard (non-robust) scenario. Nevertheless, we can still leverage the "almost" convex nature of the loss to establish the stability of adversarial training.

The following lemma gives a relationship between stability and generalization which is useful in both standard adversarial training as well as gradient descent with Moreau's envelope. When the robust training loss $\widehat{L}_{rob}(\mathbf{W}_T; S)$ is small, Lemma 4.3 provides a tighter bound than directly applying Equation (3). See Proposition A.3 for both results.

**Lemma 4.3.** (Restatement of Proposition A.3) The robust test loss satisfies the following:

$$\mathbb{E}_{S \sim \mathcal{D}^n} L_{rob}(\mathbf{W}_T) \leq \mathbb{E}_{S \sim \mathcal{D}^n} \frac{1}{1 - C_x \cdot \sup_{S \simeq S'} \delta_{\mathcal{A}}(S, S')} \widehat{L}_{rob}(\mathbf{W}_T; S).$$

This result gives a way to bound the expected robust loss. Say you want to bound the expected robust test loss by $(1 + \epsilon)$ times the expected training loss. Then, to ensure $\frac{1}{1 - \alpha_1(\eta, T)} \leq 1 + \epsilon$, we need $\alpha_1(\eta, T) \leq \frac{\epsilon}{1 + \epsilon} = O(\epsilon)$. Since $\alpha_1(\eta, T) = O(\eta \sqrt{T} + \frac{\eta T}{n} + \sqrt{\beta_1 \eta T})$, we can set different parameters in more than one way to ensure that $\alpha_1(\eta, T) = O(\epsilon)$. We can set $\beta_1 = O(\epsilon^2)$, $n = \Theta(1/\epsilon)$, $T = \Theta(1/\epsilon^2)$, $\eta = O(\frac{1}{T})$; or set $\beta_1 = O(\epsilon^3)$, $n = \Theta(1/\epsilon^2)$, $T = \Theta(1/\epsilon^4)$, $\eta = O(\frac{\epsilon}{\sqrt{T}})$.

## 4.1 Generalization Guarantees for Gradient-Based Adversarial Training

The stability guarantee we give in the following Theorem 4.4 is similar to the result in the convex case [Xing et al., 2021a]. While [Xing et al., 2021a] use the monotone subgradient condition of the convex functions, we show that the subgradients of an "almost" convex loss function are "almost" monotone. We do incur an additional term of $\exp\left(2HC_x^2 \eta T / \sqrt{m}\right)$, which is small for over-parameterized neural networks ($m \geq \text{ploy}(\eta T)$).

**Theorem 4.4.** (Restatement of Theorem 3.1) Let $S$ and $S'$ be any two neighboring data sets, i.e., they differ only in one example. Let $\mathbf{W}_T$ and $\mathbf{W}'_T$ denote the weight matrices returned after $T$ iterations of Algorithm 1 with GD on $S$ and $S'$, respectively. Then, we have

$$\|\mathbf{W}_T - \mathbf{W}'_T\|_F^2 \leq \exp\left(1 + \frac{2HC_x^2 \eta T}{\sqrt{m}}\right) \cdot \left(4C_x^2 \eta^2 (T + 1) + \frac{4C_x^2 \eta^2 (T + 1)^2}{n^2}\right).$$

We next provide an intermediate lemma that lead us to Theorem 3.2.

**Lemma 4.5.** (Restatement of Theorem B.2) Set $k = \left(1 + \frac{HC_x^2 \eta}{\sqrt{m}}\right)^{-1}$. Then after $T \leq \frac{\sqrt{m}}{HC_x^2 \eta} - 1$ iterations of Algorithm 1 with GD,

$$\frac{1}{\sum_{t=0}^{T} k^t} \sum_{t=0}^{T} k^t \widehat{L}_{rob}(\mathbf{W}_t; S) \leq \Delta_S^{\text{oracle}} + \frac{C_x^2 \eta}{2}.$$

Richards and Kuzborskij [2021] (see Lemma 2 in their paper) give an optimization guarantee by providing an upper bound on the averaged training loss $\frac{1}{T} \sum_{t=1}^{T} \widehat{L}(\mathbf{W}_t; S)$ of all iterates. In Lemma 4.5 we use a more refined analysis by considering the weighted average of the training loss. Specifically, for any weight matrix $\mathbf{W}$, we follow the standard technique in the convex case and upper bound the following:

$$\|\mathbf{W} - \mathbf{W}_{t+1}\|_F^2 = \|\mathbf{W} - \mathbf{W}_t\|_F^2 + \eta^2 \|\nabla_{\mathbf{W}} \widehat{L}_{rob}(\mathbf{W}_t; S)\|_F^2 + 2\eta \left\langle \nabla_{\mathbf{W}} \widehat{L}_{rob}(\mathbf{W}_t; S), \mathbf{W} - \mathbf{W}_t \right\rangle.$$

The second term on the right hand side is bounded by the Lipschitzness of the logistic loss. The inner product in the third term is bounded by $\widehat{L}_{rob}(\mathbf{W}; S) - \widehat{L}_{rob}(\mathbf{W}_t; S) + \frac{HC_x^2}{2\sqrt{m}}\|\mathbf{W} - \mathbf{W}_t\|_F^2$ using Lemma A.4. We finish the proof by telescoping. The weighted telescoping technique removes all of the $\|\mathbf{W} - \mathbf{W}_t\|_F^2$ terms ($t > 0$) in the upper bound, thereby giving a simpler result. The term $C_x^2 \eta / 2$ in the upper bound stems from the non-smoothness of the robust loss, and is unavoidable even if the robust loss is convex. Finally, Theorem 3.2 follows from Theorem 4.4 and Lemmas 4.3 and 4.5.

## 4.2 Generalization Guarantees for Gradient-Descent on Moreau's Envelope

Below we give the key lemmas for bounding the generalization error of GD with Moreau's envelope. The proof technique here is similar to that for standard adversarial training (in the previous section), except that we get to utilize the smoothness of Moreau's envelope. Specifically, Lemma 4.6 leverages the fact that the gradient is "almost" co-coercive to control the uniform argument stability.

**Theorem 4.6.** (Restatement of Theorem C.4) Let $S \simeq S'$ be any two neighboring data sets, i.e., $S$ and $S'$ differ only in one example. For any $\eta \leq \min\{\mu, \frac{\sqrt{m}}{8HC_x^2}\} \leq \frac{\sqrt{m}}{2HC_x^2}$, let $\mathrm{W}_T$ and $\mathrm{W}_T'$ be the weight matrices obtained by $T$ iterations of gradient descent with Moreau's envelopes on datasets $S$ and $S'$, respectively. Then, we have that

$$\|\mathrm{W}_T - \mathrm{W}_T'\|_F^2 \leq \exp\left(1 + \frac{8HC_x^2 \eta T}{\sqrt{m}}\right) \cdot \frac{16C_x^2 \eta^2 (T+1)^2}{n^2}.$$

Lemma 4.7 also leverages smoothness due to Moreau's envelope and yields a bound that does not involve the additional term $C_x^2 \eta / 2$ compared with Lemma 4.5.

**Lemma 4.7.** (Restatement of Theorem C.6) Set $k = \left(1 + \frac{2HC_x^2 \eta}{\sqrt{m}}\right)^{-1}$. After $T$ iterations of Algorithm 1 with Moreau Envelope with $\eta \leq \mu \leq \frac{\sqrt{m}}{2HC_x^2}$ and $T \leq \frac{\sqrt{m}}{HC_x^2 \eta}$, we have

$$\frac{1}{\sum\limits_{t=1}^{T} k^t} \sum_{t=1}^{T} k^t M^\mu(\mathrm{W}_t; S) \leq \Delta_S^{\mathrm{oracle}}.$$

Theorem 3.6 is naturally derived via Theorem 4.6, Lemma 4.3 and 4.7.

## 5 Conclusion

In this work, we establish the generalization guarantees for variants of adversarial training applied to two-layer networks with smooth activation functions. For over-parameterized neural networks, we present robust generalization bound that are controlled by the Adversarial Regularized Empirical Risk Minimization (ARERM) oracle, applicable to any given initialization and any data distributions. One future direction is to extend our analysis to deep neural networks and beyond neural networks with smooth activation functions.

## Acknowledgments and Disclosure of Funding

This research was supported, in part, by the DARPA GARD award HR00112020004, NSF CAREER award IIS-1943251, funding from the Institute for Assured Autonomy (IAA) at JHU, and the Spring'22 workshop on "Learning and Games" at the Simons Institute for the Theory of Computing.

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

# Supplementary Material

## A   Technical Theorems and Lemmas

**Lemma A.1.** Let $\ell(z) = \ln(1 + e^{-z})$ be the logistic loss function. We have $|\ell'(z)| \leq \min\{1, \ell(z)\}$.

*Proof of Lemma A.1.*

$$|\ell'(z)| = -\ell'(z) = \frac{1}{1 + e^z} \leq \begin{cases} 1; & (e^z > 0) \\ \ln(1 + e^{-z}). & \left(\frac{x}{1+x} \leq \ln(1+x)\right) \end{cases}$$

$\square$

**Lemma A.2.** For any sample $z = (x, y)$, and any weight matrices $W$ and $W'$, we have

$$\ell_{rob}(z, W) - \ell_{rob}(z, W') \leq C_x \|W - W'\|_2 \cdot \min\{1, \ell_{rob}(z, W)\}.$$

*Proof of Lemma A.2.*

$$\begin{aligned}
&\ell_{rob}(z, W) - \ell_{rob}(z, W') \\
&= \max_{\tilde{x} \in \mathcal{B}(x)} \ell(yf(\tilde{x}; W)) - \max_{\tilde{x} \in \mathcal{B}(x)} \ell(yf(\tilde{x}; W')) \\
&\leq \max_{\tilde{x} \in \mathcal{B}(x)} \left(\ell(yf(\tilde{x}; W)) - \ell(yf(\tilde{x}; W'))\right) \\
&\leq \max_{\tilde{x} \in \mathcal{B}(x)} |\ell'(yf(\tilde{x}; W)) \cdot (yf(\tilde{x}; W) - yf(\tilde{x}; W'))| && (\ell \text{ is convex}) \\
&= \max_{\tilde{x} \in \mathcal{B}(x)} \left| \ell'(yf(\tilde{x}; W)) \cdot \left( \sum_{s=1}^{m} a_s \phi(\langle w_s, \tilde{x}\rangle) - \sum_{s=1}^{m} a_s \phi(\langle w'_s, \tilde{x}\rangle) \right) \right| \\
&\leq \max_{\tilde{x} \in \mathcal{B}(x)} \left| \ell'(yf(\tilde{x}; W)) \cdot \frac{1}{\sqrt{m}} \left( \sum_{s=1}^{m} |\langle w_s - w'_s, \tilde{x}\rangle| \right) \right| && (\phi \text{ is 1-Lip}) \\
&\leq \max_{\tilde{x} \in \mathcal{B}(x)} \left| \ell'(yf(\tilde{x}; W)) \cdot \sqrt{\sum_{s=1}^{m} \langle w_s - w'_s, \tilde{x}\rangle^2} \right| && (\text{Cauchy's inequality}) \\
&\leq \max_{\tilde{x} \in \mathcal{B}(x)} |\ell'(yf(\tilde{x}; W)) \cdot \|W - W'\|_2 \cdot \|\tilde{x}\|_2| \\
&\leq C_x \|W - W'\|_2 \cdot \max_{\tilde{x} \in \mathcal{B}(x)} |\ell'(yf(\tilde{x}; W))| \\
&\leq C_x \|W - W'\|_2 \cdot \min\{1, \max_{\tilde{x} \in \mathcal{B}(x)} \ell(yf(\tilde{x}; W))\} && (\text{Lemma A.1}) \\
&= C_x \|W - W'\|_2 \cdot \min\{1, \ell_{rob}(z, W)\}.
\end{aligned}$$

$\square$

In the following proposition, we build the relationship between the generalization gap and uniform stability.

**Proposition A.3.** Let $S$ and $S'$ be any two neighboring data sets that differ only in one example. Let $W_t = \mathcal{A}(S), W'_t = \mathcal{A}(S')$ be the weight returned after running algorithm $\mathcal{A}$ for $t$ steps using $S$ and $S'$, respectively. $\delta_{\mathcal{A}}(S, S') = \|\mathcal{A}(S) - \mathcal{A}(S')\|_F$. Then if $\sup_{S \simeq S'} \delta_{\mathcal{A}}(S, S') < \frac{1}{C_x}$, we have

$$\mathbb{E}_{S \sim \mathcal{D}^n} L_{rob}(W_t) \leq \mathbb{E}_{S \sim \mathcal{D}^n} \frac{1}{1 - C_x \cdot \sup_{S \simeq S'} \delta_{\mathcal{A}}(S, S')} \widehat{L}_{rob}(W_t; S) \tag{4}$$

and

$$\mathbb{E}_{S \sim \mathcal{D}^n} L_{rob}(W_t) \leq \mathbb{E}_{S \sim \mathcal{D}^n} \widehat{L}_{rob}(W_t; S) + C_x \cdot \sup_{S \simeq S'} \delta_{\mathcal{A}}(S, S'). \tag{5}$$

*Proof of Proposition A.3.* Let $S$ and $S'$ differ in one example, and $z' = S'\backslash S$.

$$\mathbb{E}_{S\sim\mathcal{D}^n}\left(L_{rob}(\mathbf{W}_t) - \widehat{L}_{rob}(\mathbf{W}_t; S)\right) = \mathbb{E}_{S\cup\{z'\}\sim\mathcal{D}^{n+1}}\left[(\ell_{rob}(z', \mathbf{W}_t) - \ell_{rob}(z', \mathbf{W}_t'))\right]. \quad (6)$$

Combining Lemma A.2 and Equation (6) we get

$$\mathbb{E}_{S\sim\mathcal{D}^n}\left(L_{rob}(\mathbf{W}_t) - \widehat{L}_{rob}(\mathbf{W}_t; S)\right) \leq \mathbb{E}_{S\cup\{z'\}\sim\mathcal{D}^{n+1}}\left[C_x\|\mathbf{W}_t - \mathbf{W}_t'\|_2 \cdot \min\{1, \ell_{rob}(z', \mathbf{W}_t)\}\right].$$

Based on the definition of $\delta_{\mathcal{A}}$,

$$\mathbb{E}_{S\sim\mathcal{D}^n}\left(L_{rob}(\mathbf{W}_t) - \widehat{L}_{rob}(\mathbf{W}_t; S)\right)$$
$$\leq C_x \cdot \sup_{S\simeq S'} \delta_{\mathcal{A}}(S, S') \cdot \mathbb{E}_{S\cup\{z'\}\sim\mathcal{D}^{n+1}} \min\{1, \ell_{rob}(z', \mathbf{W}_t)\}$$
$$\leq C_x \cdot \sup_{S\simeq S'} \delta_{\mathcal{A}}(S, S') \cdot \min\{1, \mathbb{E}_{S\sim\mathcal{D}^n} L_{rob}(\mathbf{W}_t)\}.$$

Simplifying this inequality, we get

$$\mathbb{E}_{S\sim\mathcal{D}^n} L_{rob}(\mathbf{W}_t) \leq \mathbb{E}_{S\sim\mathcal{D}^n} \frac{1}{1 - C_x \cdot \sup_{S\simeq S'} \delta_{\mathcal{A}}(S, S')} \widehat{L}_{rob}(\mathbf{W}_t; S)$$

and

$$\mathbb{E}_{S\sim\mathcal{D}^n} L_{rob}(\mathbf{W}_t) \leq \mathbb{E}_{S\sim\mathcal{D}^n} \widehat{L}_{rob}(\mathbf{W}_t; S) + C_x \cdot \sup_{S\simeq S'} \delta_{\mathcal{A}}(S, S').$$

$\square$

The following lemma gives the weakly convex property of the robust loss (by considering the special case of $\beta_1 = 0$).

**Lemma A.4.** Given any data $(\mathbf{x}, y)$, for model with weight $\mathbf{W}$, let $\tilde{\mathbf{x}}(\mathbf{W}) \in \mathcal{B}(\mathbf{x})$ be an $\beta_1$-optimal adversarial examples such that $\ell(yf(\tilde{\mathbf{x}}(\mathbf{W}), \mathbf{W})) \geq \max_{\tilde{\mathbf{x}}\in\mathcal{B}(\mathbf{x})} \ell(yf(\tilde{\mathbf{x}}, \mathbf{W})) - \beta_1$. Then for any two weight matrices $\mathbf{W}^1, \mathbf{W}^2 \in \mathbb{R}^{d\times m}$, we have

$$\ell((\tilde{\mathbf{x}}(\mathbf{W}^2), y), \mathbf{W}^2) \geq \ell((\tilde{\mathbf{x}}(\mathbf{W}^1), y), \mathbf{W}^1) + \left\langle\nabla_{\mathbf{W}}\ell((\tilde{\mathbf{x}}(\mathbf{W}^1), y), \mathbf{W}^1), \mathbf{W}^2 - \mathbf{W}^1\right\rangle - \beta_1 - \frac{HC_x^2}{2\sqrt{m}}\|\mathbf{W}^2 - \mathbf{W}^1\|_2^2.$$

*Proof of Lemma A.4.*

$$\ell((\tilde{\mathbf{x}}(\mathbf{W}^2), y), \mathbf{W}^2) - \ell((\tilde{\mathbf{x}}(\mathbf{W}^1), y), \mathbf{W}^1) - \left\langle\nabla_{\mathbf{W}}\ell((\tilde{\mathbf{x}}(\mathbf{W}^1), y), \mathbf{W}^1), \mathbf{W}^2 - \mathbf{W}^1\right\rangle + \beta_1$$
$$= \ell(yf_{\mathbf{W}^2}(\tilde{\mathbf{x}}(\mathbf{W}^2))) - \ell(yf_{\mathbf{W}^1}(\tilde{\mathbf{x}}(\mathbf{W}^1))) - \left\langle\nabla_{\mathbf{W}}\ell((\tilde{\mathbf{x}}(\mathbf{W}^1), y), \mathbf{W}^1), \mathbf{W}^2 - \mathbf{W}^1\right\rangle + \beta_1$$
$$\geq \max_{\tilde{\mathbf{x}}\in\mathcal{B}(\mathbf{x})} \ell(yf_{\mathbf{W}^2}(\tilde{\mathbf{x}})) - \ell(yf_{\mathbf{W}^1}(\tilde{\mathbf{x}}(\mathbf{W}^1))) - \left\langle\nabla_{\mathbf{W}}\ell((\tilde{\mathbf{x}}(\mathbf{W}^1), y), \mathbf{W}^1), \mathbf{W}^2 - \mathbf{W}^1\right\rangle$$

(By definition of $\beta_1$-optimal adversarial examples)

$$\geq \ell(yf_{\mathbf{W}^2}(\tilde{\mathbf{x}}(\mathbf{W}^1))) - \ell(yf_{\mathbf{W}^1}(\tilde{\mathbf{x}}(\mathbf{W}^1))) - \left\langle\nabla_{\mathbf{W}}\ell((\tilde{\mathbf{x}}(\mathbf{W}^1), y), \mathbf{W}^1), \mathbf{W}^2 - \mathbf{W}^1\right\rangle$$
$$\geq \ell'(yf_{\mathbf{W}^1}(\tilde{\mathbf{x}}(\mathbf{W}^1))) \cdot \left(yf_{\mathbf{W}^2}(\tilde{\mathbf{x}}(\mathbf{W}^1)) - yf_{\mathbf{W}^1}(\tilde{\mathbf{x}}(\mathbf{W}^1))\right) - \left\langle\nabla_{\mathbf{W}}\ell((\tilde{\mathbf{x}}(\mathbf{W}^1), y), \mathbf{W}^1), \mathbf{W}^2 - \mathbf{W}^1\right\rangle$$

($\ell$ is convex)

$$= \ell'(yf_{\mathbf{W}^1}(\tilde{\mathbf{x}}(\mathbf{W}^1)))y\sum_{s=1}^m a_s \left(\phi(\langle\mathbf{w}_s^2, \tilde{\mathbf{x}}(\mathbf{W}^1)\rangle) - \phi(\langle\mathbf{w}_s^1, \tilde{\mathbf{x}}(\mathbf{W}^1)\rangle) - \phi'(\langle\mathbf{w}_s^1, \tilde{\mathbf{x}}(\mathbf{W}^1)\rangle)\langle\mathbf{w}_s^2 - \mathbf{w}_s^1, \tilde{\mathbf{x}}(\mathbf{W}^1)\rangle\right)$$

$$\geq -|\ell'(yf_{\mathbf{W}^1}(\tilde{\mathbf{x}}(\mathbf{W}^1)))|\sum_{s=1}^m \frac{1}{\sqrt{m}} \cdot \frac{H}{2}\left\langle\mathbf{w}_s^2 - \mathbf{w}_s^1, \tilde{\mathbf{x}}(\mathbf{W}^1)\right\rangle^2$$

($\phi$ is $H$-smooth)

$$\geq -1 \cdot \frac{H}{2\sqrt{m}}\|(\mathbf{W}^2 - \mathbf{W}^1)^\mathsf{T}\tilde{\mathbf{x}}(\mathbf{W}^1)\|_2^2$$

(Lemma A.1)

$$\geq -\frac{HC_x^2}{2\sqrt{m}}\|\mathbf{W}^2 - \mathbf{W}^1\|_2^2.$$

($\|\tilde{\mathbf{x}}(\mathbf{W}^1)\|_2 \leq C_x$)

$\square$

The following lemma tells us the gradient has a universal upper bound.

**Lemma A.5.** For any data $(\mathrm{x}, y)$ and any weight matrix $\mathrm{W}$,

$$\|\nabla_{\mathrm{W}}\ell(yf(\mathrm{x};\mathrm{W}))\|_F \leq C_x.$$

*Proof of Lemma A.5.* Since $\nabla_{\mathrm{W}}\ell(yf(\mathrm{x};\mathrm{W})) = [\ell'(yf(\mathrm{x};\mathrm{W}))ya_s\phi'(\langle\mathrm{w}_s,\mathrm{x}\rangle)\mathrm{x}]_{s=1}^m$,

$$
\begin{aligned}
\|\nabla_{\mathrm{W}}\ell(yf(\mathrm{x};\mathrm{W}))\|_F &= \sqrt{\sum_{s=1}^m \|\ell'(yf(\mathrm{x};\mathrm{W}))ya_s\phi'(\langle\mathrm{W}_s,\mathrm{x}\rangle)\mathrm{x}\|_2^2} \\
&\leq -C_x \cdot \ell'(yf(\mathrm{x};\mathrm{W})) && (|a_s| = \tfrac{1}{\sqrt{m}}, \phi' \leq 1, \|\mathrm{x}\|_2 \leq C_x) \\
&\leq C_x. && (\text{Lemma A.1})
\end{aligned}
$$

$\square$

## B  Missing Proofs in Section 3.1

Now we give the uniform argument stability upper bound.

**Theorem B.1.** (Restatement of Theorem 3.1) Let $S$ and $S^{(i)}$ only differ in the $i$-th data. $\mathrm{W}_T$ and $\mathrm{W}_T^{(i)}$ denote the weight matrices returned after after running Algorithm GD for $T$ iterations on $S$ and $S^{(i)}$, respectively. Then we have

$$\|\mathrm{W}_T - \mathrm{W}_T^{(i)}\|_F^2 \leq e^{1+\frac{2HC_x^2\eta T}{\sqrt{m}}}\left(4C_x^2\eta^2(T+1) + \frac{4C_x^2\eta^2(T+1)^2}{n^2} + 4\beta_1\eta(T+1)\right).$$

*Proof of Theorem B.1.* For any weight matrix $\mathrm{W}$ and any perturbation of the $j$-th data $\tilde{\mathrm{x}}_j \in \mathcal{B}(\mathrm{x}_j)$ $(j \neq i)$, define $L_{S\setminus i}(\mathrm{W}; \{\tilde{\mathrm{x}}_j\}_{j\neq i}) = \frac{1}{n}\sum_{j\neq i}\ell(y_jf(\tilde{\mathrm{x}}_j;\mathrm{W}))$ to be the loss of the (perturbed) data set without including the $i$-th data. Let $\tilde{\mathrm{x}}_j(\mathrm{W}_t)$ denote the $\beta_1$-optimal adversarial example of $\mathrm{x}_j$ given $\mathrm{W}_t$, and $\tilde{\mathrm{x}}_j(\mathrm{W}_t^{(i)})$ denote the $\beta_1$-optimal adversarial example of $\mathrm{x}_j$ given $\mathrm{W}_t^{(i)}$. We first show that the gradient is an "almost" monotone operator, which is derived from the weakly convex property of the robust loss (see Lemma A.4).

$$
\begin{aligned}
&\beta_1 + \frac{HC_x^2}{2\sqrt{m}}\|\mathrm{W}_t - \mathrm{W}_t^{(i)}\|_F^2 + L_{S\setminus i}(\mathrm{W}_t; \{\tilde{\mathrm{x}}_j(\mathrm{W}_t)\}_{j\neq i}) \\
\geq& \frac{1}{n}\sum_{j\neq i}\left[\beta_1 + \frac{HC_x^2}{2\sqrt{m}}\|\mathrm{W}_t - \mathrm{W}_t^{(i)}\|_F^2 + \ell(y_jf(\tilde{\mathrm{x}}_j(\mathrm{W}_t);\mathrm{W}_t))\right] \\
\geq& \frac{1}{n}\sum_{j\neq i}\left[\ell(y_jf(\tilde{\mathrm{x}}_j(\mathrm{W}_t^{(i)});\mathrm{W}_t^{(i)})) + \left\langle\nabla_{\mathrm{W}}\ell(y_jf(\tilde{\mathrm{x}}_j(\mathrm{W}_t^{(i)});\mathrm{W}_t^{(i)})), \mathrm{W}_t - \mathrm{W}_t^{(i)}\right\rangle\right] && (\text{Lemma A.4}) \\
=& L_{S\setminus i}(\mathrm{W}_t^{(i)}; \{\tilde{\mathrm{x}}_j(\mathrm{W}_t^{(i)})\}_{j\neq i}) + \left\langle\nabla_{\mathrm{W}}L_{S\setminus i}(\mathrm{W}_t^{(i)}; \{\tilde{\mathrm{x}}_j(\mathrm{W}_t^{(i)})\}_{j\neq i}), \mathrm{W}_t - \mathrm{W}_t^{(i)}\right\rangle.
\end{aligned}
$$

Similarly, we get

$$
\begin{aligned}
&\beta_1 + \frac{HC_x^2}{2\sqrt{m}}\|\mathrm{W}_t - \mathrm{W}_t^{(i)}\|_F^2 + L_{S\setminus i}(\mathrm{W}_t^{(i)}; \{\tilde{\mathrm{x}}_j(\mathrm{W}_t^{(i)})\}_{j\neq i}) \\
\geq& L_{S\setminus i}(\mathrm{W}_t; \{\tilde{\mathrm{x}}_j(\mathrm{W}_t)\}_{j\neq i}) + \left\langle\nabla_{\mathrm{W}}L_{S\setminus i}(\mathrm{W}_t; \{\tilde{\mathrm{x}}_j(\mathrm{W}_t)\}_{j\neq i}), \mathrm{W}_t^{(i)} - \mathrm{W}_t\right\rangle.
\end{aligned}
$$

Adding these two inequalities together, we get

$$
\begin{aligned}
&\left\langle\nabla_{\mathrm{W}}L_{S\setminus i}(\mathrm{W}_t^{(i)}; \{\tilde{\mathrm{x}}_j(\mathrm{W}_t^{(i)})\}_{j\neq i}) - \nabla_{\mathrm{W}}L_{S\setminus i}(\mathrm{W}_t; \{\tilde{\mathrm{x}}_j(\mathrm{W}_t)\}_{j\neq i}), \mathrm{W}_t^{(i)} - \mathrm{W}_t\right\rangle \\
\geq& -2\beta_1 - \frac{HC_x^2}{\sqrt{m}}\|\mathrm{W}_t - \mathrm{W}_t^{(i)}\|_F^2. && (7)
\end{aligned}
$$

Now we start upper bounding $\|\mathbf{W}_t - \mathbf{W}_t^{(i)}\|_F$.

$$\|\mathbf{W}_{t+1} - \mathbf{W}_{t+1}^{(i)}\|_F^2$$

$$=\|\mathbf{W}_t - \eta\nabla_\mathbf{w} L_{S\setminus i}(\mathbf{W}_t; \{\tilde{\mathbf{x}}_j(\mathbf{W}_t)\}_{j\neq i}) - \frac{\eta}{n}\nabla_\mathbf{w}\ell(y_i f(\tilde{\mathbf{x}}_i(\mathbf{W}_t); \mathbf{W}_t))$$
$$- \mathbf{W}_t^{(i)} + \eta\nabla_\mathbf{w} L_{S\setminus i}(\mathbf{W}_t^{(i)}; \{\tilde{\mathbf{x}}_j(\mathbf{W}_t^{(i)})\}_{j\neq i}) + \frac{\eta}{n}\nabla_\mathbf{w}\ell(y_i' f(\tilde{\mathbf{x}}_i'(\mathbf{W}_t^{(i)}); \mathbf{W}_t^{(i)}))\|_F^2$$

$$\leq \left( \|\mathbf{W}_t - \eta\nabla_\mathbf{w} L_{S\setminus i}(\mathbf{W}_t; \{\tilde{\mathbf{x}}_j(\mathbf{W}_t)\}_{j\neq i}) - \mathbf{W}_t^{(i)} + \eta\nabla_\mathbf{w} L_{S\setminus i}(\mathbf{W}_t^{(i)}; \{\tilde{\mathbf{x}}_j(\mathbf{W}_t^{(i)})\}_{j\neq i})\|_F + \frac{2\eta C_x}{n} \right)^2$$
$$\text{(Lemma A.5)}$$

$$\leq \frac{T+2}{T+1}\|\mathbf{W}_t - \eta\nabla_\mathbf{w} L_{S\setminus i}(\mathbf{W}_t; \{\tilde{\mathbf{x}}_j(\mathbf{W}_t)\}_{j\neq i}) - \mathbf{W}_t^{(i)} + \eta\nabla_\mathbf{w} L_{S\setminus i}(\mathbf{W}_t^{(i)}; \{\tilde{\mathbf{x}}_j(\mathbf{W}_t^{(i)})\}_{j\neq i})\|_F^2$$
$$+ (T+2)\frac{4\eta^2 C_x^2}{n^2} \qquad\qquad ((a+b)^2 \leq (1+p)a^2 + (1+1/p)b^2 \text{ for } p > 0.)$$

$$=\frac{T+2}{T+1}\|\mathbf{W}_t - \mathbf{W}_t^{(i)}\|_F^2 + \frac{T+2}{T+1}\|\eta\nabla_\mathbf{w} L_{S\setminus i}(\mathbf{W}_t; \{\tilde{\mathbf{x}}_j(\mathbf{W}_t)\}_{j\neq i}) - \eta\nabla_\mathbf{w} L_{S\setminus i}(\mathbf{W}_t^{(i)}; \{\tilde{\mathbf{x}}_j(\mathbf{W}_t^{(i)})\}_{j\neq i})\|_F^2$$
$$- 2\eta\frac{T+2}{T+1}\left\langle \nabla_\mathbf{w} L_{S\setminus i}(\mathbf{W}_t^{(i)}; \{\tilde{\mathbf{x}}_j(\mathbf{W}_t^{(i)})\}_{j\neq i}) - \nabla_\mathbf{w} L_{S\setminus i}(\mathbf{W}_t; \{\tilde{\mathbf{x}}_j(\mathbf{W}_t)\}_{j\neq i}), \mathbf{W}_t^{(i)} - \mathbf{W}_t \right\rangle$$
$$+ (T+2)\frac{4\eta^2 C_x^2}{n^2}$$

$$\leq \frac{T+2}{T+1}\|\mathbf{W}_t - \mathbf{W}_t^{(i)}\|_F^2 + \frac{T+2}{T+1}4\eta^2 C_x^2 + \frac{T+2}{T+1}(4\beta_1\eta + \frac{2HC_x^2}{\sqrt{m}}\eta\|\mathbf{W}_t - \mathbf{W}_t^{(i)}\|_F^2) + (T+2)\frac{4\eta^2 C_x^2}{n^2}$$
$$\text{(Lemma A.5 and Equation (7))}$$

$$=\frac{T+2}{T+1}\left( (1 + \frac{2HC_x^2}{\sqrt{m}}\eta)\|\mathbf{W}_t - \mathbf{W}_t^{(i)}\|_F^2 + 4\eta^2 C_x^2 + 4\beta_1\eta + (T+1)\frac{4\eta^2 C_x^2}{n^2} \right).$$

Define $\gamma = \frac{T+2}{T+1}(1 + \frac{2HC_x^2}{\sqrt{m}}\eta)$, then the inequality above can be written as

$$\|\mathbf{W}_{t+1} - \mathbf{W}_{t+1}^{(i)}\|_F^2 \leq \gamma\|\mathbf{W}_t - \mathbf{W}_t^{(i)}\|_F^2 + \frac{T+2}{T+1}\left( 4\eta^2 C_x^2 + 4\beta_1\eta + (T+1)\frac{4\eta^2 C_x^2}{n^2} \right).$$

Dividing both sides by $\gamma^{t+1}$,

$$\frac{\|\mathbf{W}_{t+1} - \mathbf{W}_{t+1}^{(i)}\|_F^2}{\gamma^{t+1}} \leq \frac{\|\mathbf{W}_t - \mathbf{W}_t^{(i)}\|_F^2}{\gamma^t} + \frac{T+2}{T+1}\frac{4\eta^2 C_x^2 + 4\beta_1\eta + (T+1)\frac{4\eta^2 C_x^2}{n^2}}{\gamma^{t+1}}.$$

Summing up this inequality for $t = 0, 1, \ldots, T-1$, we obtain

$$\|\mathbf{W}_T - \mathbf{W}_T^{(i)}\|_F^2 \leq \gamma^T \sum_{t=0}^{T-1} \frac{T+2}{T+1}\frac{4\eta^2 C_x^2 + 4\beta_1\eta + (T+1)\frac{4\eta^2 C_x^2}{n^2}}{\gamma^{t+1}}$$

$$\leq \frac{\gamma^T}{\gamma - 1}\frac{T+2}{T+1}\left( 4\eta^2 C_x^2 + 4\beta_1\eta + (T+1)\frac{4\eta^2 C_x^2}{n^2} \right)$$

$$\leq (T+1)\gamma^T \frac{T+2}{T+1}\left( 4\eta^2 C_x^2 + 4\beta_1\eta + (T+1)\frac{4\eta^2 C_x^2}{n^2} \right)$$

$$= (T+1)(\frac{T+2}{T+1})^{T+1}(1 + \frac{2HC_x^2}{\sqrt{m}}\eta)^T \left( 4\eta^2 C_x^2 + 4\beta_1\eta + (T+1)\frac{4\eta^2 C_x^2}{n^2} \right)$$

$$\leq (T+1)e \cdot e^{\frac{2HC_x^2\eta T}{\sqrt{m}}} \left( 4\eta^2 C_x^2 + 4\beta_1\eta + (T+1)\frac{4\eta^2 C_x^2}{n^2} \right) \qquad (1 + x \leq e^x)$$

$$= e^{1 + \frac{2HC_x^2\eta T}{\sqrt{m}}} \left( 4C_x^2\eta^2(T+1) + \frac{4C_x^2\eta^2(T+1)^2}{n^2} + 4\beta_1\eta(T+1) \right).$$

$$\qquad\qquad\qquad\qquad\qquad\qquad\qquad\qquad\qquad\qquad\qquad\qquad\qquad\qquad\qquad \square$$

The proof of Theorem 3.1 is immediately obtained from Theorem B.1 by observing $e^{1+\frac{2HC_x^2\eta T}{\sqrt{m}}} \le e^3$.

Next we give an optimization guarantee. We show that when $T$ is sufficiently large, the robust training loss can approach the adversarial regularized empirical risk minimization oracle. We do not need early stopping in the Theorem below.

**Theorem B.2.** After running Algorithm GD for $T$ iterations, we have

$$\min_{0 \le t \le T} \widehat{L}_{rob}(\mathbf{W}_t; S) \le \min_{\mathbf{W} \in \mathbb{R}^{d \times m}} \left( \widehat{L}_{rob}(\mathbf{W}; S) + \frac{HC_x^2}{2\sqrt{m}(1 - \frac{1}{(1+\frac{HC_x^2\eta}{\sqrt{m}})^{T+1}})} \|\mathbf{W} - \mathbf{W}_0\|_F^2 + \frac{C_x^2\eta}{2} + \beta_1 \right).$$

*Proof of Theorem B.2.* For any given $\mathbf{W}$, we have

$$\begin{aligned}
&\|\mathbf{W} - \mathbf{W}_{t+1}\|_F^2 \\
=& \|\mathbf{W} - \mathbf{W}_t + \eta \nabla_{\mathbf{W}} L(\mathbf{W}_t; \{\tilde{\mathbf{x}}_i(\mathbf{W}_t)\}_{i=1}^n)\|_F^2 \\
=& \|\mathbf{W} - \mathbf{W}_t\|_F^2 + \eta^2 \|\nabla_{\mathbf{W}} L(\mathbf{W}_t; \{\tilde{\mathbf{x}}_i(\mathbf{W}_t)\}_{i=1}^n)\|_F^2 + 2\eta \langle \nabla_{\mathbf{W}} L(\mathbf{W}_t; \{\tilde{\mathbf{x}}_i(\mathbf{W}_t)\}_{i=1}^n), \mathbf{W} - \mathbf{W}_t \rangle \\
\le& \|\mathbf{W} - \mathbf{W}_t\|_F^2 + \eta^2 C_x^2 + \frac{HC_x^2\eta}{\sqrt{m}} \|\mathbf{W} - \mathbf{W}_t\|_F^2 + 2\eta \widehat{L}_{rob}(\mathbf{W}; S) - 2\eta L(\mathbf{W}_t; \{\tilde{\mathbf{x}}_i(\mathbf{W}_t)\}_{i=1}^n) \\
&\qquad\qquad\qquad\qquad\qquad\qquad\qquad\qquad\qquad\qquad\qquad\qquad \text{(Lemma A.5 and Lemma A.4)} \\
=& (1 + \frac{HC_x^2\eta}{\sqrt{m}}) \|\mathbf{W} - \mathbf{W}_t\|_F^2 + \eta^2 C_x^2 + 2\eta \widehat{L}_{rob}(\mathbf{W}; S) - 2\eta L(\mathbf{W}_t; \{\tilde{\mathbf{x}}_i(\mathbf{W}_t)\}_{i=1}^n).
\end{aligned}$$

Dividing both sides by $(1 + \frac{HC_x^2\eta}{\sqrt{m}})^{t+1}$ we get

$$\frac{\|\mathbf{W} - \mathbf{W}_{t+1}\|_F^2}{(1 + \frac{HC_x^2\eta}{\sqrt{m}})^{t+1}} \le \frac{\|\mathbf{W} - \mathbf{W}_t\|_F^2}{(1 + \frac{HC_x^2\eta}{\sqrt{m}})^t} + \frac{\eta^2 C_x^2}{(1 + \frac{HC_x^2\eta}{\sqrt{m}})^{t+1}} + \frac{2\eta(\widehat{L}_{rob}(\mathbf{W}; S) - L(\mathbf{W}_t; \{\tilde{\mathbf{x}}_i(\mathbf{W}_t)\}_{i=1}^n))}{(1 + \frac{HC_x^2\eta}{\sqrt{m}})^{t+1}}. \tag{8}$$

Taking the sum for $t = 0, 1, \ldots, T$ we get

$$\begin{aligned}
&2\eta \sum_{t=0}^T \frac{\widehat{L}_{rob}(\mathbf{W}; S)}{(1 + \frac{HC_x^2\eta}{\sqrt{m}})^{t+1}} + \|\mathbf{W} - \mathbf{W}_0\|_F^2 + \sum_{t=0}^T \frac{\eta^2 C_x^2}{(1 + \frac{HC_x^2\eta}{\sqrt{m}})^{t+1}} \\
\ge& 2\eta \sum_{t=0}^T \frac{L(\mathbf{W}_t; \{\tilde{\mathbf{x}}_i(\mathbf{W}_t)\}_{i=1}^n)}{(1 + \frac{HC_x^2\eta}{\sqrt{m}})^{t+1}} \\
\ge& 2\eta \sum_{t=0}^T \frac{\widehat{L}_{rob}(\mathbf{W}_t; S) - \beta_1}{(1 + \frac{HC_x^2\eta}{\sqrt{m}})^{t+1}} \\
\ge& 2\eta \min_{0 \le t \le T} \widehat{L}_{rob}(\mathbf{W}_t; S) \cdot \sum_{t=0}^T \frac{1}{(1 + \frac{HC_x^2\eta}{\sqrt{m}})^{t+1}} - 2\eta \sum_{t=0}^T \frac{\beta_1}{(1 + \frac{HC_x^2\eta}{\sqrt{m}})^{t+1}}.
\end{aligned} \tag{9}$$

Simplifying the above inequality, we have

$$\min_{0 \le t \le T} \widehat{L}_{rob}(\mathbf{W}_t; S) \le \widehat{L}_{rob}(\mathbf{W}; S) + \frac{HC_x^2}{2\sqrt{m}(1 - \frac{1}{(1+\frac{HC_x^2\eta}{\sqrt{m}})^{T+1}})} \|\mathbf{W} - \mathbf{W}_0\|_F^2 + \frac{C_x^2\eta}{2} + \beta_1.$$

$\square$

**Theorem 3.2.** Define $\alpha_1(\eta, T) := \mathcal{O}(C_x^2 \eta \sqrt{T} + C_x^2 \frac{\eta T}{n} + C_x \sqrt{\beta_1 \eta T})$. Assume that the width of the networks satisfies $m \ge H^2 C_x^4 \eta^2 (T+1)^2$, and $\alpha_1(\eta, T) < 1$. Then, after $T$ iterations of Algorithm 1 with GD, we have

$$\min_{\lceil \frac{9T}{10} \rceil \le t \le T} \mathbb{E}_{S \sim \mathcal{D}^n} \varepsilon_{gen}(\mathbf{W}_t) \le \frac{17\alpha_1(\eta, T)}{1 - \alpha_1(\eta, T)} \left[ \mathbb{E}_{S \sim \mathcal{D}^n} \Delta_S^{\text{oracle}} + \frac{C_x^2\eta}{2} + \beta_1 \right],$$

and

$$\min_{0 \le t \le T} \mathbb{E}_{S \sim \mathcal{D}^n} L_{rob}(\mathbf{W}_t) \le \frac{1}{1 - \alpha_1(\eta, T)} \left[ \mathbb{E}_{S \sim \mathcal{D}^n} \Delta_S^{\text{oracle}} + \frac{C_x^2\eta}{2} + \beta_1 \right].$$

*Proof of Theorem 3.2.* Define $t_0 := \lceil \frac{9T}{10} \rceil$ and $k := \frac{1}{1+\frac{HC_x^2\eta}{\sqrt{m}}}$. From equation (9), we have

$$2\eta \sum_{t=0}^{T} \frac{\widehat{L}_{rob}(\mathbf{W}; S)}{(1+\frac{HC_x^2\eta}{\sqrt{m}})^{t+1}} + \|\mathbf{W} - \mathbf{W}_0\|_F^2 + \sum_{t=0}^{T} \frac{\eta^2 C_x^2}{(1+\frac{HC_x^2\eta}{\sqrt{m}})^{t+1}}$$

$$\geq 2\eta \sum_{t=t_0}^{T} \frac{L(\mathbf{W}_t; \{\tilde{\mathbf{x}}_i(\mathbf{W}_t)\}_{i=1}^n)}{(1+\frac{HC_x^2\eta}{\sqrt{m}})^{t+1}}$$

$$\geq 2\eta \sum_{t=t_0}^{T} \frac{\widehat{L}_{rob}(\mathbf{W}_t; S) - \beta_1}{(1+\frac{HC_x^2\eta}{\sqrt{m}})^{t+1}}.$$

Taking minimum over all weight matrices $\mathbf{W}$,

$$\sum_{t=t_0}^{T} k^{t+1} \left( \widehat{L}_{rob}(\mathbf{W}_t; S) - \beta_1 \right)$$

$$\leq \min_{\mathbf{W}} \left( \sum_{t=0}^{T} k^{t+1} \widehat{L}_{rob}(\mathbf{W}; S) + \frac{\|\mathbf{W} - \mathbf{W}_0\|_F^2}{2\eta} + \sum_{t=0}^{T} \frac{\eta C_x^2}{2} k^{t+1} \right)$$

$$= \sum_{t=0}^{T} k^{t+1} \cdot \min_{\mathbf{W}} \left( \widehat{L}_{rob}(\mathbf{W}; S) + \frac{\|\mathbf{W} - \mathbf{W}_0\|_F^2}{2\eta \sum_{t=0}^{T} k^{t+1}} + \frac{C_x^2 \eta}{2} \right)$$

$$\leq \sum_{t=0}^{T} k^{t+1} \cdot \min_{\mathbf{W}} \left( \widehat{L}_{rob}(\mathbf{W}; S) + \frac{\|\mathbf{W} - \mathbf{W}_0\|_F^2}{\eta(T+1)} + \frac{C_x^2 \eta}{2} \right) \qquad (m \geq H^2 C_x^4 \eta^2 (T+1)^2)$$

$$\leq \sum_{t=0}^{T} k^{t+1} \cdot \left( \Delta_S^{\text{oracle}} + \frac{C_x^2 \eta}{2} \right).$$

Taking the expectation on both sides, we get

$$\sum_{t=t_0}^{T} k^{t+1} \left( \min_{t_0 \leq t \leq T} \mathbb{E}_{S \sim \mathcal{D}^n} \widehat{L}_{rob}(\mathbf{W}_t; S) - \beta_1 \right) \leq \sum_{t=t_0}^{T} k^{t+1} \left( \mathbb{E}_{S \sim \mathcal{D}^n} \widehat{L}_{rob}(\mathbf{W}_t; S) - \beta_1 \right)$$

$$\leq \sum_{t=0}^{T} k^{t+1} \cdot \left( \mathbb{E}_{S \sim \mathcal{D}^n} \Delta_S^{\text{oracle}} + \frac{C_x^2 \eta}{2} \right).$$

Simplifying the equation above, we get

$$\min_{t_0 \leq t \leq T} \mathbb{E}_{S \sim \mathcal{D}^n} \widehat{L}_{rob}(\mathbf{W}_t; S) \leq \beta_1 + \frac{\sum_{t=0}^{T} k^{t+1}}{\sum_{t=t_0}^{T} k^{t+1}} \cdot \left( \mathbb{E}_{S \sim \mathcal{D}^n} \Delta_S^{\text{oracle}} + \frac{C_x^2 \eta}{2} \right) \qquad (10)$$

$$\leq \beta_1 + \left( \sum_{r=0}^{9} \left( \frac{1}{k} \right)^{r \frac{T+1}{10}} \right) \cdot \left( \mathbb{E}_{S \sim \mathcal{D}^n} \Delta_S^{\text{oracle}} + \frac{C_x^2 \eta}{2} \right)$$

$$\leq \beta_1 + \left( \sum_{r=0}^{9} e^{\frac{r}{10}} \right) \cdot \left( \mathbb{E}_{S \sim \mathcal{D}^n} \Delta_S^{\text{oracle}} + \frac{C_x^2 \eta}{2} \right)$$

$$\qquad (m \geq H^2 C_x^4 \eta^2 (T+1)^2)$$

$$\leq \beta_1 + 17 \left( \mathbb{E}_{S \sim \mathcal{D}^n} \Delta_S^{\text{oracle}} + \frac{C_x^2 \eta}{2} \right).$$

Equation (4) gives us for any $t \leq T$, $\mathbb{E}_{S \sim \mathcal{D}^n} \varepsilon_{gen}(\mathbf{W}_t) \leq \frac{\alpha_1(\eta,T)}{1-\alpha_1(\eta,T)} \mathbb{E}_{S \sim \mathcal{D}^n} \widehat{L}_{rob}(\mathbf{W}_t; S)$. Therefore,

$$\min_{t_0 \leq t \leq T} \mathbb{E}_{S \sim \mathcal{D}^n} \varepsilon_{gen}(\mathbf{W}_t) \leq \frac{\alpha_1(\eta,T)}{1-\alpha_1(\eta,T)} \min_{t_0 \leq t \leq T} \mathbb{E}_{S \sim \mathcal{D}^n} \widehat{L}_{rob}(\mathbf{W}_t; S)$$
$$\leq \frac{\alpha_1(\eta,T)}{1-\alpha_1(\eta,T)} \left( \beta_1 + 17 \left( \mathbb{E}_{S \sim \mathcal{D}^n} \Delta_S^{\text{oracle}} + \frac{C_x^2 \eta}{2} \right) \right).$$

The proof of the second statement takes a similar approach. Following the same procedure, we can replace $t_0$ by 0 in equation (10), and get

$$\min_{0 \leq t \leq T} \mathbb{E}_{S \sim \mathcal{D}^n} \widehat{L}_{rob}(\mathbf{W}_t; S) \leq \mathbb{E}_{S \sim \mathcal{D}^n} \Delta_S^{\text{oracle}} + \frac{C_x^2 \eta}{2} + \beta_1.$$

Combining with equation (4),

$$\min_{0 \leq t \leq T} \mathbb{E}_{S \sim \mathcal{D}^n} L_{rob}(\mathbf{W}_t) \leq \frac{1}{1-\alpha_1(\eta,T)} \min_{0 \leq t \leq T} \mathbb{E}_{S \sim \mathcal{D}^n} \widehat{L}_{rob}(\mathbf{W}_t; S)$$
$$\leq \frac{1}{1-\alpha_1(\eta,T)} \left( \mathbb{E}_{S \sim \mathcal{D}^n} \Delta_S^{\text{oracle}} + \frac{C_x^2 \eta}{2} + \beta_1 \right).$$

$\square$

**Corollary 3.3.** After $T \leq \mathcal{O}(\min\{n^2, \frac{1}{\beta_1^2}\})$ iterations of Algorithm 1 with GD using a step size of $\eta = \Theta(\frac{1}{C_x^2 \sqrt{T}})$ on a network with width $m \geq \Omega(T)$, for any weight matrix $\mathbf{W}$

$$\min_{0 \leq t \leq T} \mathbb{E}_{S \sim \mathcal{D}^n} L_{rob}(\mathbf{W}_t) \leq 1.1 L_{rob}(\mathbf{W}) + \mathcal{O}\left( \frac{C_x^2 \|\mathbf{W} - \mathbf{W}_0\|_F^2}{\sqrt{T}} \right) + \mathcal{O}\left( \frac{1}{\sqrt{T}} \right).$$

*Proof of Corollary 3.3.* Under the conditions of the corollary, we have $m \geq H^2 C_x^4 \eta^2 (T+1)^2$, and $\alpha_1(\eta,T) = \mathcal{O}(C_x^2 \eta \sqrt{T} + C_x^2 \frac{\eta T}{n} + C_x \sqrt{\beta_1 \eta T})$ can be small enough so that $\frac{1}{1-\alpha_1(\eta,T)} \leq 1.1$. Then it is clear that this corollary is a special case of Theorem 3.2. $\square$

We now extend the previous ideas to stochastic adversarial training.

**Lemma B.3.** After $T$ iterations of Algorithm 1 with SGD, for any weight matrix $\mathbf{W}$,

$$\min_{0 \leq t \leq T} (\ell_{rob}(z_{t+1}, \mathbf{W}_t) - \ell_{rob}(z_{t+1}, \mathbf{W})) \leq \frac{HC_x^2}{2\sqrt{m}(1 - \frac{1}{(1 + \frac{HC_x^2 \eta}{\sqrt{m}})^{T+1}})} \|\mathbf{W} - \mathbf{W}_0\|_F^2 + \frac{C_x^2 \eta}{2} + \beta_1.$$

*Proof of Lemma B.3.* The proof proceeds similarly as Theorem B.2.

$\|\mathbf{W} - \mathbf{W}_{t+1}\|_F^2$
$= \|\mathbf{W} - \mathbf{W}_t + \eta \nabla_{\mathbf{W}} \ell((\tilde{x}_{t+1}(\mathbf{W}_t), y_{t+1}), \mathbf{W}_t)\|_F^2$
$= \|\mathbf{W} - \mathbf{W}_t\|_F^2 + \eta^2 \|\nabla_{\mathbf{W}} \ell((\tilde{x}_{t+1}(\mathbf{W}_t), y_{t+1}), \mathbf{W}_t)\|_F^2 + 2\eta \langle \nabla_{\mathbf{W}} \ell((\tilde{x}_{t+1}(\mathbf{W}_t), y_{t+1}), \mathbf{W}_t), \mathbf{W} - \mathbf{W}_t \rangle$
$\leq \|\mathbf{W} - \mathbf{W}_t\|_F^2 + \eta^2 C_x^2 + \frac{HC_x^2 \eta}{\sqrt{m}} \|\mathbf{W} - \mathbf{W}_t\|_F^2 + 2\eta \ell_{rob}(z_{t+1}, \mathbf{W}) - 2\eta \ell((\tilde{x}_{t+1}(\mathbf{W}_t), y_{t+1}), \mathbf{W}_t)$

(Lemma A.5 and Lemma A.4)

$\leq (1 + \frac{HC_x^2 \eta}{\sqrt{m}}) \|\mathbf{W} - \mathbf{W}_t\|_F^2 + \eta^2 C_x^2 + 2\eta \ell_{rob}(z_{t+1}, \mathbf{W}) - 2\eta \ell_{rob}(z_{t+1}, \mathbf{W}_t) + 2\eta \beta_1.$

Dividing both sides by $(1 + \frac{HC_x^2 \eta}{\sqrt{m}})^{t+1}$, we get

$$\frac{\|\mathbf{W} - \mathbf{W}_{t+1}\|_F^2}{(1 + \frac{HC_x^2 \eta}{\sqrt{m}})^{t+1}} \leq \frac{\|\mathbf{W} - \mathbf{W}_t\|_F^2}{(1 + \frac{HC_x^2 \eta}{\sqrt{m}})^t} + \frac{\eta^2 C_x^2 + 2\eta \ell_{rob}(z_{t+1}, \mathbf{W}) - 2\eta \ell_{rob}(z_{t+1}, \mathbf{W}_t) + 2\eta \beta_1}{(1 + \frac{HC_x^2 \eta}{\sqrt{m}})^{t+1}}.$$

Taking the sum of the above equation for $t = 0, 1, \ldots, T$:

$$
\begin{aligned}
&\|\mathbf{W} - \mathbf{W}_0\|_F^2 + \sum_{t=0}^{T} \frac{\eta^2 C_x^2 + 2\eta \beta_1}{(1 + \frac{HC_x^2 \eta}{\sqrt{m}})^{t+1}} \\
&\geq 2\eta \sum_{t=0}^{T} \frac{\ell_{rob}(\mathbf{z}_{t+1}, \mathbf{W}_t) - \ell_{rob}(\mathbf{z}_{t+1}, \mathbf{W})}{(1 + \frac{HC_x^2 \eta}{\sqrt{m}})^{t+1}} \\
&\geq 2\eta \sum_{t=0}^{T} \frac{\min_{0 \leq t \leq T} (\ell_{rob}(\mathbf{z}_{t+1}, \mathbf{W}_t) - \ell_{rob}(\mathbf{z}_{t+1}, \mathbf{W}))}{(1 + \frac{HC_x^2 \eta}{\sqrt{m}})^{t+1}}.
\end{aligned}
\tag{11}
$$

Simplifying the above inequality we have

$$
\min_{0 \leq t \leq T} (\ell_{rob}(\mathbf{z}_{t+1}, \mathbf{W}_t) - \ell_{rob}(\mathbf{z}_{t+1}, \mathbf{W})) \leq \frac{HC_x^2}{2\sqrt{m}(1 - \frac{1}{(1+\frac{HC_x^2 \eta}{\sqrt{m}})^{T+1}})} \|\mathbf{W} - \mathbf{W}_0\|_F^2 + \frac{C_x^2 \eta}{2} + \beta_1.
$$

$\square$

**Theorem 3.4.** After $T$ iterations of Algorithm 1 with SGD on a network of width $m \geq H^2 C_x^4 \eta^2 (T + 1)^2$ we have that for any weight matrix W,

$$
\min_{0 \leq t \leq T} \mathbb{E}_{\{z_1, \ldots, z_t\} \sim \mathcal{D}^t} L_{rob}(\mathbf{W}_t) \leq L_{rob}(\mathbf{W}) + \frac{\|\mathbf{W} - \mathbf{W}_0\|_F^2}{\eta(T+1)} + \frac{C_x^2 \eta}{2} + \beta_1.
$$

*Proof of Theorem 3.4.* Taking the expectation over $S \sim \mathcal{D}^n$ on both sides of Equation (11), we obtain

$$
\begin{aligned}
&\|\mathbf{W} - \mathbf{W}_0\|_F^2 + \sum_{t=0}^{T} \frac{\eta^2 C_x^2 + 2\eta \beta}{(1 + \frac{HC_x^2 \eta}{\sqrt{m}})^{t+1}} \\
&\geq 2\eta \sum_{t=0}^{T} \frac{\mathbb{E}_{\{z_1, \ldots, z_t\} \sim \mathcal{D}^t} \mathbb{E}_{z_{t+1} \sim \mathcal{D}} \ell_{rob}(\mathbf{z}_{t+1}, \mathbf{W}_t) - \mathbb{E}_{z_{t+1} \sim \mathcal{D}} \ell_{rob}(\mathbf{z}_{t+1}, \mathbf{W})}{(1 + \frac{HC_x^2 \eta}{\sqrt{m}})^{t+1}} \\
&= 2\eta \sum_{t=0}^{T} \frac{\mathbb{E}_{\{z_1, \ldots, z_t\} \sim \mathcal{D}^t} L_{rob}(\mathbf{W}_t) - L_{rob}(\mathbf{W})}{(1 + \frac{HC_x^2 \eta}{\sqrt{m}})^{t+1}} \\
&\geq 2\eta \sum_{t=0}^{T} \frac{\min_{0 \leq t \leq T} \mathbb{E}_{\{z_1, \ldots, z_t\} \sim \mathcal{D}^t} L_{rob}(\mathbf{W}_t) - L_{rob}(\mathbf{W})}{(1 + \frac{HC_x^2 \eta}{\sqrt{m}})^{t+1}}.
\end{aligned}
$$

Simplifying the above inequality, we get

$$
\begin{aligned}
\min_{0 \leq t \leq T} \mathbb{E}_{\{z_1, \ldots, z_t\} \sim \mathcal{D}^t} L_{rob}(\mathbf{W}_t) &\leq L_{rob}(\mathbf{W}) + \frac{HC_x^2}{2\sqrt{m}(1 - \frac{1}{(1+\frac{HC_x^2 \eta}{\sqrt{m}})^{T+1}})} \|\mathbf{W} - \mathbf{W}_0\|_F^2 + \frac{C_x^2 \eta}{2} + \beta_1 \\
&\leq L_{rob}(\mathbf{W}) + \frac{\|\mathbf{W} - \mathbf{W}_0\|_F^2}{\eta(T+1)} + \frac{C_x^2 \eta}{2} + \beta_1.
\end{aligned}
$$

$$(m \geq H^2 C_x^4 \eta^2 (T+1)^2)$$

$\square$

## C  Missing Proofs in Section 3.2

From Lemma A.4, for any $\mu < \frac{\sqrt{m}}{HC_x^2}$, $L_{rob}(\mathbf{U}) + \frac{1}{2\mu}\|\mathbf{U} - \mathbf{W}\|_F^2$ is strongly convex in U. Recall that the Moreau envelope is defined as

$$
M^\mu(\mathbf{W}; S) = \min_{\mathbf{U}} \left( \widehat{L}_{rob}(\mathbf{U}; S) + \frac{1}{2\mu}\|\mathbf{U} - \mathbf{W}\|_F^2 \right).
$$

The minimizer of the optimization problem above is denoted as

$$\mathrm{U}^{\mu}(\mathrm{W}; S) = \underset{\mathrm{U}}{\arg\min} \left( \widehat{L}_{rob}(\mathrm{U}; S) + \frac{1}{2\mu} \|\mathrm{U} - \mathrm{W}\|_F^2 \right).$$

We borrow a few properties of the Moreau envelope from Xiao et al. [2024].

**Lemma C.1.** For any $\mu < \frac{\sqrt{m}}{HC_x^2}$,

1. $\min_{\mathrm{W}} M^{\mu}(\mathrm{W}; S)$ has the same global solution set as $\min_{\mathrm{W}} \widehat{L}_{rob}(\mathrm{W}; S)$.

2. The gradient of $M^{\mu}(\mathrm{W}; S)$ is $\nabla_{\mathrm{W}} M^{\mu}(\mathrm{W}; S) = \frac{1}{\mu} (\mathrm{W} - \mathrm{U}^{\mu}(\mathrm{W}; S))$.

3. $M^{\mu}(\mathrm{W}; S) + \frac{\|\mathrm{W}\|_F^2}{2\left(\frac{\sqrt{m}}{HC_x^2} - \mu\right)}$ is convex.

4. $\mathrm{U}^{\mu}(\mathrm{W}; S)$ is $\frac{\frac{\sqrt{m}}{HC_x^2}}{\frac{\sqrt{m}}{HC_x^2} - \mu}$-Lipschitz in W w.r.t the Frobenius norm.

5. $M^{\mu}(\mathrm{W}; S)$ is $\max\left\{ \frac{1}{\mu}, \frac{1}{\frac{\sqrt{m}}{HC_x^2} - \mu} \right\}$-smooth.

6. $\widehat{L}_{rob}(\mathrm{W}; S) - \frac{C_x^2}{2\left(\frac{1}{\mu} - \frac{HC_x^2}{\sqrt{m}}\right)} \leq M^{\mu}(\mathrm{W}; S) \leq \widehat{L}_{rob}(\mathrm{W}; S).$

7. $\|\nabla_{\mathrm{W}} M^{\mu}(\mathrm{W}; S)\|_F \leq C_x.$

*Proof of Lemma C.1.* The first 5 statements are covered in the proof of [Xiao et al., 2024, Lemma A.1]. For the statement 6,

$$\widehat{L}_{rob}(\mathrm{W}; S) = \widehat{L}_{rob}(\mathrm{W}; S) + \frac{1}{2\mu} \|\mathrm{W} - \mathrm{W}\|_F^2$$

$$\geq \min_{\mathrm{U}} \left( \widehat{L}_{rob}(\mathrm{U}; S) + \frac{1}{2\mu} \|\mathrm{U} - \mathrm{W}\|_F^2 \right)$$

$$= M^{\mu}(\mathrm{W}; S)$$

$$= \widehat{L}_{rob}(\mathrm{W}; S) + \min_{\mathrm{U}} \left( \widehat{L}_{rob}(\mathrm{U}; S) - \widehat{L}_{rob}(\mathrm{W}; S) + \frac{1}{2\mu} \|\mathrm{U} - \mathrm{W}\|_F^2 \right)$$

$$\geq \widehat{L}_{rob}(\mathrm{W}; S) + \min_{\mathrm{U}} \left( \left\langle \nabla_{\mathrm{W}} \widehat{L}_{rob}(\mathrm{W}; S), \mathrm{U} - \mathrm{W} \right\rangle - \frac{HC_x^2}{2\sqrt{m}} \|\mathrm{U} - \mathrm{W}\|_F^2 + \frac{1}{2\mu} \|\mathrm{U} - \mathrm{W}\|_F^2 \right)$$

("almost" convex robust loss from Lemma A.4)

$$\geq \widehat{L}_{rob}(\mathrm{W}; S) + \min_{\mathrm{U}} \left( -C_x \|\mathrm{U} - \mathrm{W}\|_F - \frac{HC_x^2}{2\sqrt{m}} \|\mathrm{U} - \mathrm{W}\|_F^2 + \frac{1}{2\mu} \|\mathrm{U} - \mathrm{W}\|_F^2 \right)$$

(Lemma A.5)

$$= \widehat{L}_{rob}(\mathrm{W}; S) - \frac{C_x^2}{2\left(\frac{1}{\mu} - \frac{HC_x^2}{\sqrt{m}}\right)}.$$

Now we prove statement 7. For any $\gamma \in (0, 1)$, from the definition of the Moreau envelope,

$$\widehat{L}_{rob}(\mathrm{U}^{\mu}(\mathrm{W}; S); S) + \frac{1}{2\mu} \|\mathrm{U}^{\mu}(\mathrm{W}; S) - \mathrm{W}\|_F^2$$

$$\leq \widehat{L}_{rob}((1 - \gamma)\mathrm{W} + \gamma \mathrm{U}^{\mu}(\mathrm{W}; S); S) + \frac{1}{2\mu} \|(1 - \gamma)\mathrm{W} + \gamma \mathrm{U}^{\mu}(\mathrm{W}; S) - \mathrm{W}\|_F^2$$

($\mathrm{U}^{\mu}(\mathrm{W}; S)$ obtains the minimum)

$$\leq \widehat{L}_{rob}(\mathrm{U}^{\mu}(\mathrm{W}; S); S) + C_x(1 - \gamma)\|\mathrm{U}^{\mu}(\mathrm{W}; S) - \mathrm{W}\|_F + \frac{\gamma^2}{2\mu} \|\mathrm{U}^{\mu}(\mathrm{W}; S) - \mathrm{W}\|_F^2.$$

(The robust loss is $C_x$-Lip from Lemma A.2)

Simplifying the inequality above, we get

$$\|U^\mu(W; S) - W\|_F \le \frac{2\mu C_x}{1 + \gamma}.$$

Let $\gamma \to 1$, we get

$$\|\nabla_W M^\mu(W; S)\|_F = \frac{1}{\mu}\|U^\mu(W; S) - W\|_F \le C_x.$$

$\square$

Next we show a result similar as [Xiao et al., 2024, Lemma A.2]. They use the first-order optimal condition to prove the result, which only holds for the smooth loss functions. Here we give a different proof that doesn't depend on the subgradient, so it can be applied to the robust loss.

**Lemma C.2.** Let $\tilde{U}^\mu(W; S)$ be any $\beta_2$-optimal minimizer of $\min_U \left( \widehat{L}_{rob}(U; S) + \frac{1}{2\mu}\|U - W\|_F^2 \right)$. For two data sets $S$ and $S^{(i)}$ that differ in only one example and any weight matrix W, we have

$$\|U^\mu(W; S) - U^\mu(W; S^{(i)})\|_F \le \frac{2C_x}{n\left(\frac{1}{\mu} - \frac{HC_x^2}{\sqrt{m}}\right)}$$

and

$$\|\tilde{U}^\mu(W; S) - U^\mu(W; S)\|_F \le \sqrt{\frac{2\beta_2}{\frac{1}{\mu} - \frac{HC_x^2}{\sqrt{m}}}}.$$

*Proof of Lemma C.2.* From strong convexity of the regularized robust loss, we have

$$\widehat{L}_{rob}(U^\mu(W; S^{(i)}); S) + \frac{1}{2\mu}\|U^\mu(W; S^{(i)}) - W\|_F^2$$

$$\ge \widehat{L}_{rob}(U^\mu(W; S); S) + \frac{1}{2\mu}\|U^\mu(W; S) - W\|_F^2 + (\frac{1}{2\mu} - \frac{HC_x^2}{2\sqrt{m}})\|U^\mu(W; S^{(i)}) - U^\mu(W; S)\|_F^2,$$

and similarly,

$$\widehat{L}_{rob}(U^\mu(W; S); S^{(i)}) + \frac{1}{2\mu}\|U^\mu(W; S) - W\|_F^2$$

$$\ge \widehat{L}_{rob}(U^\mu(W; S^{(i)}); S^{(i)}) + \frac{1}{2\mu}\|U^\mu(W; S^{(i)}) - W\|_F^2 + (\frac{1}{2\mu} - \frac{HC_x^2}{2\sqrt{m}})\|U^\mu(W; S) - U^\mu(W; S^{(i)})\|_F^2.$$

Adding these two inequalities,

$$(\frac{1}{\mu} - \frac{HC_x^2}{\sqrt{m}})\|U^\mu(W; S) - U^\mu(W; S^{(i)})\|_F^2$$

$$\le \left[\widehat{L}_{rob}(U^\mu(W; S^{(i)}); S) - \widehat{L}_{rob}(U^\mu(W; S^{(i)}); S^{(i)})\right] + \left[\widehat{L}_{rob}(U^\mu(W; S); S^{(i)}) - \widehat{L}_{rob}(U^\mu(W; S); S)\right]$$

$$= \frac{1}{n}\left[\ell_{rob}(z_i, U^\mu(W; S^{(i)})) - \ell_{rob}(z_i', U^\mu(W; S^{(i)}))\right] + \frac{1}{n}[\ell_{rob}(z_i', U^\mu(W; S)) - \ell_{rob}(z_i, U^\mu(W; S))]$$

$$= \frac{1}{n}\left[\ell_{rob}(z_i, U^\mu(W; S^{(i)})) - \ell_{rob}(z_i, U^\mu(W; S))\right] + \frac{1}{n}\left[\ell_{rob}(z_i', U^\mu(W; S)) - \ell_{rob}(z_i', U^\mu(W; S^{(i)}))\right]$$

$$\le \frac{2C_x}{n}\|U^\mu(W; S) - U^\mu(W; S^{(i)})\|_F. \tag{Lemma A.2}$$

Therefore,

$$\|U^\mu(W; S) - U^\mu(W; S^{(i)})\|_F \le \frac{2C_x}{n\left(\frac{1}{\mu} - \frac{HC_x^2}{\sqrt{m}}\right)}. \tag{12}$$

From the strong convexity of the regularized robust loss, we also have

$$\beta_2 \geq \left( \widehat{L}_{rob}(\tilde{U}^\mu(W; S); S) + \frac{1}{2\mu} \|\tilde{U}^\mu(W; S) - W\|_F^2 \right)$$

$$- \left( \widehat{L}_{rob}(U^\mu(W; S); S) + \frac{1}{2\mu} \|U^\mu(W; S) - W\|_F^2 \right)$$

$$\geq (\frac{1}{2\mu} - \frac{HC_x^2}{2\sqrt{m}}) \|\tilde{U}^\mu(W; S) - U^\mu(W; S)\|_F^2.$$

We get $\|\tilde{U}^\mu(W; S) - U^\mu(W; S)\|_F \leq \sqrt{\frac{\beta_2}{\frac{1}{2\mu} - \frac{HC_x^2}{2\sqrt{m}}}}.$ $\qquad\square$

Now we show an upper bound that is key to bounding the stability of the weight matrix.

**Lemma C.3.** For any $\eta \leq \mu \leq \frac{\sqrt{m}}{2HC_x^2}$ and any weight matrices $W_1$ and $W_2$,

$$\|W^1 - \eta \nabla_W M^\mu(W^1; S) - W^2 + \eta \nabla_W M^\mu(W^2; S)\|_F^2 \leq \frac{\|W^1 - W^2\|_F^2}{1 - \frac{4HC_x^2 \eta}{\sqrt{m}}}.$$

*Proof of Lemma C.3.* Define $\psi_1(W) = M^\mu(W; S) - M^\mu(W^1; S) - \langle \nabla_W M^\mu(W^1; S), W - W^1 \rangle$. From Lemma C.1, $\psi_1(W)$ is $\frac{1}{\mu}$-smooth. From the standard descent lemma of smooth functions, for any $\eta \leq \mu$,

$$\psi_1(W^2 - \eta \nabla_W \psi_1(W^2)) \leq \psi_1(W^2) - \frac{\eta}{2} \|\nabla_W \psi_1(W^2)\|_F^2$$

$$= \psi_1(W^2) - \frac{\eta}{2} \|\nabla_W M^\mu(W^2; S) - \nabla_W M^\mu(W^1; S)\|_F^2.$$

Since $\psi_1(W) + \frac{\|W\|_F^2}{2\left(\frac{\sqrt{m}}{HC_x^2} - \mu\right)}$ is convex from Lemma C.1,

$$\psi_1(W^2 - \eta \nabla_W \psi_1(W^2))$$

$$\geq \psi_1(W^1) + \langle \nabla \psi_1(W^1), W^2 - \eta \nabla_W \psi_1(W^2) - W^1 \rangle - \frac{\|W^2 - \eta \nabla_W \psi_1(W^2) - W^1\|_F^2}{2\left(\frac{\sqrt{m}}{HC_x^2} - \mu\right)}$$

$$= \psi_1(W^1) - \frac{\|W^2 - \eta \nabla_W \psi_1(W^2) - W^1\|_F^2}{2\left(\frac{\sqrt{m}}{HC_x^2} - \mu\right)}$$

$$\geq \psi_1(W^1) - \frac{\|W^2 - \eta \nabla_W M^\mu(W^2; S) - W^1 + \eta \nabla_W M^\mu(W^1; S)\|_F^2}{\frac{\sqrt{m}}{HC_x^2}}.$$

Combining the two inequalities above,

$$M^\mu(W^2; S) - M^\mu(W^1; S) - \langle \nabla_W M^\mu(W^1; S), W^2 - W^1 \rangle$$

$$= \psi_1(W^2) - \psi_1(W^1)$$

$$\geq \frac{\eta}{2} \|\nabla_W M^\mu(W^2; S) - \nabla_W M^\mu(W^1; S)\|_F^2 - \frac{\|W^2 - \eta \nabla_W M^\mu(W^2; S) - W^1 + \eta \nabla_W M^\mu(W^1; S)\|_F^2}{\frac{\sqrt{m}}{HC_x^2}}.$$

Similarly, we can get the counterpart of this equation:

$$M^\mu(W^1; S) - M^\mu(W^2; S) - \langle \nabla_W M^\mu(W^2; S), W^1 - W^2 \rangle$$

$$\geq \frac{\eta}{2} \|\nabla_W M^\mu(W^1; S) - \nabla_W M^\mu(W^2; S)\|_F^2 - \frac{\|W^1 - \eta \nabla_W M^\mu(W^1; S) - W^2 + \eta \nabla_W M^\mu(W^2; S)\|_F^2}{\frac{\sqrt{m}}{HC_x^2}}.$$

Adding these two inequalities, we get

$$\left\langle \nabla_W M^\mu(W^2; S) - \nabla_W M^\mu(W^1; S), W^2 - W^1 \right\rangle$$

$$\geq \eta \|\nabla_W M^\mu(W^1; S) - \nabla_W M^\mu(W^2; S)\|_F^2 - \frac{2\|W^1 - \eta\nabla_W M^\mu(W^1; S) - W^2 + \eta\nabla_W M^\mu(W^2; S)\|_F^2}{\frac{\sqrt{m}}{HC_x^2}}.$$

Thus,

$$\|W^1 - \eta\nabla_W M^\mu(W^1; S) - W^2 + \eta\nabla_W M^\mu(W^2; S)\|_F^2$$

$$= \|W^1 - W^2\|_F^2 + \eta^2 \|\nabla_W M^\mu(W^1; S) - \nabla_W M^\mu(W^2; S)\|_F^2$$
$$\quad - 2\eta \left\langle \nabla_W M^\mu(W^1; S) - \nabla_W M^\mu(W^2; S), W^1 - W^2 \right\rangle$$

$$\leq \|W^1 - W^2\|_F^2 + \eta^2 \|\nabla_W M^\mu(W^1; S) - \nabla_W M^\mu(W^2; S)\|_F^2$$
$$\quad - 2\eta^2 \|\nabla_W M^\mu(W^1; S) - \nabla_W M^\mu(W^2; S)\|_F^2$$
$$\quad + \frac{4\eta \|W^1 - \eta\nabla_W M^\mu(W^1; S) - W^2 + \eta\nabla_W M^\mu(W^2; S)\|_F^2}{\frac{\sqrt{m}}{HC_x^2}}$$

$$\leq \|W^1 - W^2\|_F^2 + \frac{4HC_x^2\eta\|W^1 - \eta\nabla_W M^\mu(W^1; S) - W^2 + \eta\nabla_W M^\mu(W^2; S)\|_F^2}{\sqrt{m}}.$$

We get the desired result by simplifying the above inequality. $\qquad\square$

**Theorem C.4.** (Restatement of Theorem 3.5) For any $\eta \leq \min\{\mu, \frac{\sqrt{m}}{8HC_x^2}\} \leq \frac{\sqrt{m}}{2HC_x^2}$, let $W_T$ and $W_T^{(i)}$ be the weight matrices returned after running Algorithm 1 with Moreau Envelope using $S$ and $S^{(i)}$ respectively for $T$ iterations. Here $S$ and $S^{(i)}$ only differ in the $i$-th data. We have

$$\|W_T - W_T^{(i)}\|_F^2 \leq e^{1 + \frac{8HC_x^2\eta T}{\sqrt{m}}}(T+1)^2 \left( \frac{4C_x\eta}{n} + 4\eta\sqrt{\frac{\beta_2}{\mu}} \right)^2.$$

*Proof of Theorem C.4.*

$$\|W_{t+1} - W_{t+1}^{(i)}\|_F^2$$

$$= \|W_t - \eta\frac{1}{\mu}(W_t - \tilde{U}^\mu(W_t; S)) - W_t^{(i)} + \eta\frac{1}{\mu}(W_t^{(i)} - \tilde{U}^\mu(W_t^{(i)}; S^{(i)}))\|_F^2$$

$$\leq \left( \|W_t - \eta\frac{1}{\mu}(W_t - U^\mu(W_t; S)) - W_t^{(i)} + \eta\frac{1}{\mu}(W_t^{(i)} - U^\mu(W_t^{(i)}; S^{(i)}))\|_F + \frac{2\eta}{\mu}\sqrt{\frac{2\beta_2}{\frac{1}{\mu} - \frac{HC_x^2}{\sqrt{m}}}} \right)^2$$
$$\hfill \text{(Lemma C.2)}$$

$$\leq \Bigg( \|W_t - \eta\frac{1}{\mu}(W_t - U^\mu(W_t; S)) - W_t^{(i)} + \eta\frac{1}{\mu}(W_t^{(i)} - U^\mu(W_t^{(i)}; S))\|_F + \frac{2C_x\eta}{\mu n\left(\frac{1}{\mu} - \frac{HC_x^2}{\sqrt{m}}\right)}$$

$$+ \frac{2\eta}{\mu}\sqrt{\frac{2\beta_2}{\frac{1}{\mu} - \frac{HC_x^2}{\sqrt{m}}}} \Bigg)^2 \hfill \text{(Lemma C.2)}$$

$$\leq \left( \|W_t - \eta\frac{1}{\mu}(W_t - U^\mu(W_t; S)) - W_t^{(i)} + \eta\frac{1}{\mu}(W_t^{(i)} - U^\mu(W_t^{(i)}; S))\|_F + \frac{4C_x\eta}{n} + 4\eta\sqrt{\frac{\beta_2}{\mu}} \right)^2$$

$$\leq \frac{T+2}{T+1}\|W_t - \eta\nabla_W M^\mu(W_t; S) - W_t^{(i)} + \eta\nabla_W M^\mu(W_t^{(i)}; S)\|_F^2 + (T+2)\left( \frac{4C_x\eta}{n} + 4\eta\sqrt{\frac{\beta_2}{\mu}} \right)^2$$
$$\hfill ((a+b)^2 \leq (1+p)a^2 + (1+1/p)b^2 \text{ for } p > 0.)$$

$$\leq \frac{T+2}{T+1} \cdot \frac{1}{1 - \frac{4HC_x^2\eta}{\sqrt{m}}}\|W_t - W_t^{(i)}\|_F^2 + (T+2)\left( \frac{4C_x\eta}{n} + 4\eta\sqrt{\frac{\beta_2}{\mu}} \right)^2. \hfill \text{(Lemma C.3)}$$

Define $\gamma = \frac{T+2}{T+1} \cdot \frac{1}{1 - \frac{4HC_x^2\eta}{\sqrt{m}}}$, then the inequality above can be written as

$$\|\mathbf{W}_{t+1} - \mathbf{W}_{t+1}^{(i)}\|_F^2 \leq \gamma\|\mathbf{W}_t - \mathbf{W}_t^{(i)}\|_F^2 + (T+2)\left(\frac{4C_x\eta}{n} + 4\eta\sqrt{\frac{\beta_2}{\mu}}\right)^2.$$

Dividing both sides by $\gamma^{t+1}$ and summing up the inequality, we get

$$\|\mathbf{W}_T - \mathbf{W}_T^{(i)}\|_F^2 \leq \frac{\gamma^T}{\gamma - 1}(T+2)\left(\frac{4C_x\eta}{n} + 4\eta\sqrt{\frac{\beta_2}{\mu}}\right)^2$$

$$\leq \gamma^T(T+1)(T+2)\left(\frac{4C_x\eta}{n} + 4\eta\sqrt{\frac{\beta_2}{\mu}}\right)^2$$

$$= \left(\frac{T+2}{T+1}\right)^{T+1}\left(\frac{1}{1 - \frac{4HC_x^2\eta}{\sqrt{m}}}\right)^T(T+1)^2\left(\frac{4C_x\eta}{n} + 4\eta\sqrt{\frac{\beta_2}{\mu}}\right)^2$$

$$\leq e^{1 + \frac{\frac{4HC_x^2\eta}{\sqrt{m}}T}{1 - \frac{4HC_x^2\eta}{\sqrt{m}}}}(T+1)^2\left(\frac{4C_x\eta}{n} + 4\eta\sqrt{\frac{\beta_2}{\mu}}\right)^2 \qquad (\frac{1}{1-x} \leq e^{\frac{x}{1-x}})$$

$$\leq e^{1 + \frac{8HC_x^2\eta T}{\sqrt{m}}}(T+1)^2\left(\frac{4C_x\eta}{n} + 4\eta\sqrt{\frac{\beta_2}{\mu}}\right)^2. \qquad (\eta \leq \frac{\sqrt{m}}{8HC_x^2})$$

$\square$

The proof of Theorem 3.5 is obvious from Theorem C.4 by observing $e^{1 + \frac{8HC_x^2\eta T}{\sqrt{m}}} \leq e^9$.

We have the following corollary by combining Theorem C.4, Lemma C.1-6 and Proposition A.3.

**Corollary C.5.** Assume the width of the networks satisfies $m \geq H^2C_x^4\eta^2T^2$. After $T$ iterations of Algorithm 1 with Moreau Envelope with $\eta \leq \min\{\mu, \frac{\sqrt{m}}{8HC_x^2}\} \leq \frac{\sqrt{m}}{2HC_x^2}$, we have

$$\mathbb{E}_{S \sim \mathcal{D}^n} L_{rob}(\mathbf{W}_T) \leq \mathbb{E}_{S \sim \mathcal{D}^n} \frac{M^\mu(\mathbf{W}_T; S) + C_x^2\mu}{1 - e^{4.5}C_x(T+1)\left(\frac{4C_x\eta}{n} + 4\eta\sqrt{\frac{\beta_2}{\mu}}\right)}$$

and

$$\mathbb{E}_{S \sim \mathcal{D}^n} L_{rob}(\mathbf{W}_T) \leq \mathbb{E}_{S \sim \mathcal{D}^n} M^\mu(\mathbf{W}_T; S) + C_x^2\mu + e^{4.5}C_x(T+1)\left(\frac{4C_x\eta}{n} + 4\eta\sqrt{\frac{\beta_2}{\mu}}\right).$$

Now we derive an optimization guarantee of optimizing the Moreau envelope.

**Theorem C.6.** Assume the width of the networks satisfies $m \geq H^2C_x^4\eta^2T^2$. After running Algorithm 1 with Moreau Envelope for $T$ iterations with $\eta \leq \mu \leq \frac{\sqrt{m}}{2HC_x^2}$, we have

$$\min_{1 \leq t \leq T} M^\mu(\mathbf{W}_t; S) \leq \min_{\mathbf{W}}\left(\widehat{L}_{rob}(\mathbf{W}; S) + \frac{2}{\eta T}\|\mathbf{W} - \mathbf{W}_0\|_F^2 + 2\eta(T+1)\frac{\beta_2}{\mu}\right).$$

*Proof of Theorem C.6.* From Lemma C.1, $M^\mu(\mathbf{W}; S)$ is $\frac{1}{\mu}$ smooth, so

$$M^\mu(\mathbf{W}_{t+1}; S)$$

$$\leq M^\mu(\mathbf{W}_t; S) + \langle \mathbf{W}_{t+1} - \mathbf{W}_t, \nabla_\mathbf{W} M^\mu(\mathbf{W}_t; S)\rangle + \frac{1}{2\mu}\|\mathbf{W}_{t+1} - \mathbf{W}_t\|_F^2. \qquad (13)$$

From the weakly convex property Lemma C.1-3,

$M^\mu(\mathbf{W}; S)$

$$\geq M^\mu(\mathbf{W}_t; S) + \langle \nabla_\mathbf{W} M^\mu(\mathbf{W}_t; S), \mathbf{W} - \mathbf{W}_t \rangle - \frac{HC_x^2}{\sqrt{m}} \|\mathbf{W}_t - \mathbf{W}\|_F^2$$

$$= M^\mu(\mathbf{W}_t; S) + \langle \nabla_\mathbf{W} M^\mu(\mathbf{W}_t; S), \mathbf{W} - \mathbf{W}_{t+1} \rangle + \langle \mathbf{W}_{t+1} - \mathbf{W}_t, \nabla_\mathbf{W} M^\mu(\mathbf{W}_t; S) \rangle - \frac{HC_x^2}{\sqrt{m}} \|\mathbf{W}_t - \mathbf{W}\|_F^2$$

$$\geq M^\mu(\mathbf{W}_{t+1}; S) + \langle \nabla_\mathbf{W} M^\mu(\mathbf{W}_t; S), \mathbf{W} - \mathbf{W}_{t+1} \rangle - \frac{1}{2\mu} \|\mathbf{W}_{t+1} - \mathbf{W}_t\|_F^2 - \frac{HC_x^2}{\sqrt{m}} \|\mathbf{W}_t - \mathbf{W}\|_F^2$$

$$\text{(equation (13))}$$

$$\geq M^\mu(\mathbf{W}_{t+1}; S) + \frac{1}{\eta} \langle \mathbf{W}_{t+1} - \mathbf{W}_t, \mathbf{W}_{t+1} - \mathbf{W} \rangle - \frac{1}{2\eta} \|\mathbf{W}_{t+1} - \mathbf{W}_t\|_F^2 - \frac{HC_x^2}{\sqrt{m}} \|\mathbf{W}_t - \mathbf{W}\|_F^2$$

$$(\mu \geq \eta)$$

$$- \frac{1}{\mu} \|\mathbf{U}^\mu(\mathbf{W}_t; S) - \tilde{\mathbf{U}}^\mu(\mathbf{W}_t; S)\|_F \cdot \|\mathbf{W}_{t+1} - \mathbf{W}\|_F \qquad \text{(Lemma C.1-2.)}$$

$$\geq M^\mu(\mathbf{W}_{t+1}; S) + \frac{1}{\eta} \langle \mathbf{W}_{t+1} - \mathbf{W}_t, \mathbf{W}_{t+1} - \mathbf{W} \rangle - \frac{1}{2\eta} \|\mathbf{W}_{t+1} - \mathbf{W}_t\|_F^2 - \frac{HC_x^2}{\sqrt{m}} \|\mathbf{W}_t - \mathbf{W}\|_F^2$$

$$- 2\sqrt{\frac{\beta_2}{\mu}} \cdot \|\mathbf{W}_{t+1} - \mathbf{W}\|_F. \qquad \text{(Lemma C.2)}$$

From the inequality above, for any weight matrix $\mathbf{W}$,

$\|\mathbf{W}_{t+1} - \mathbf{W}\|_F^2$

$$= \|\mathbf{W}_t - \mathbf{W}\|_F^2 - \|\mathbf{W}_{t+1} - \mathbf{W}_t\|_F^2 + 2 \langle \mathbf{W}_{t+1} - \mathbf{W}_t, \mathbf{W}_{t+1} - \mathbf{W} \rangle$$

$$\leq \|\mathbf{W}_t - \mathbf{W}\|_F^2 + \left( 2\eta M^\mu(\mathbf{W}; S) - 2\eta M^\mu(\mathbf{W}_{t+1}; S) + \frac{2HC_x^2\eta}{\sqrt{m}} \|\mathbf{W}_t - \mathbf{W}\|_F^2 + 4\eta\sqrt{\frac{\beta_2}{\mu}} \cdot \|\mathbf{W}_{t+1} - \mathbf{W}\|_F \right).$$

Simplifying the above inequality and combining with Lemma C.1-6 gives us

$$\|\mathbf{W}_{t+1} - \mathbf{W}\|_F^2 \leq \left( 1 + \frac{2HC_x^2\eta}{\sqrt{m}} \right) \|\mathbf{W}_t - \mathbf{W}\|_F^2 + 4\eta\sqrt{\frac{\beta_2}{\mu}} \|\mathbf{W}_{t+1} - \mathbf{W}\|_F$$

$$+ 2\eta \widehat{L}_{rob}(\mathbf{W}; S) - 2\eta M^\mu(\mathbf{W}_{t+1}; S)$$

$$\leq \left( 1 + \frac{2HC_x^2\eta}{\sqrt{m}} \right) \|\mathbf{W}_t - \mathbf{W}\|_F^2 + \frac{1}{T+1} \|\mathbf{W}_{t+1} - \mathbf{W}\|_F^2 + 4\eta^2(T+1)\frac{\beta_2}{\mu}$$

$$+ 2\eta \widehat{L}_{rob}(\mathbf{W}; S) - 2\eta M^\mu(\mathbf{W}_{t+1}; S).$$

Thus,

$$\|\mathbf{W}_{t+1} - \mathbf{W}\|_F^2 \leq \left( 1 + \frac{1}{T} \right) \left( 1 + \frac{2HC_x^2\eta}{\sqrt{m}} \right) \|\mathbf{W}_t - \mathbf{W}\|_F^2$$

$$+ 2\eta \left( 1 + \frac{1}{T} \right) \left( \widehat{L}_{rob}(\mathbf{W}; S) - M^\mu(\mathbf{W}_{t+1}; S) + 2\eta(T+1)\frac{\beta_2}{\mu} \right). \qquad (14)$$

Dividing both sides by $\left( 1 + \frac{1}{T} \right)^{t+1} \left( 1 + \frac{2HC_x^2\eta}{\sqrt{m}} \right)^{t+1}$ and summing up the inequality for $t = 0, 1, \dots, T - 1$, we get

$$\min_{1 \leq t \leq T} M^\mu(\mathbf{W}_t; S) \leq \widehat{L}_{rob}(\mathbf{W}; S) + 2\eta(T+1)\frac{\beta_2}{\mu} + \frac{\|\mathbf{W} - \mathbf{W}_0\|_F^2}{2\eta \left( 1 + \frac{1}{T} \right) \sum_{t=1}^{T} \frac{1}{\left( 1 + \frac{1}{T} \right)^t \left( 1 + \frac{2HC_x^2\eta}{\sqrt{m}} \right)^t}}$$

$$\leq \widehat{L}_{rob}(\mathbf{W}; S) + \frac{2}{\eta T} \|\mathbf{W} - \mathbf{W}_0\|_F^2 + 2\eta(T+1)\frac{\beta_2}{\mu},$$

where in the last inequality we use $\left( 1 + \frac{1}{T} \right) \left( 1 + \frac{2HC_x^2\eta}{\sqrt{m}} \right) \leq \left( 1 + \frac{1}{T} \right) \left( 1 + \frac{2}{T} \right) \leq 1 + 3\frac{T+1}{T^2}$. $\qquad \square$

**Theorem 3.6.** Define $\alpha_2(\eta, T) := \mathcal{O}(C_x^2 \frac{\eta T}{n} + C_x \eta T \sqrt{\frac{\beta_2}{\mu}})$. Assume $\alpha_2(\eta, T) < 1$. Then, after $T \geq 8$ iterations of Algorithm Moreau Envelope with step-size $\eta \leq \mu$ on a network of width $m \geq H^2 C_x^4 \eta^2 T^2$, we have

$$\min_{[\frac{9T}{10}] \leq t \leq T} \mathbb{E}_{S \sim \mathcal{D}^n} \varepsilon_{gen}(\mathbf{W}_t) \leq \frac{55\alpha_2(\eta, T)}{1 - \alpha_2(\eta, T)} \left[ \mathbb{E}_{S \sim \mathcal{D}^n} \Delta_S^{\text{oracle}} + C_x^2 \mu + 2\eta(T+1)\frac{\beta_2}{\mu} \right],$$

and

$$\min_{1 \leq t \leq T} \mathbb{E}_{S \sim \mathcal{D}^n} L_{rob}(\mathbf{W}_t) \leq \frac{1}{1 - \alpha_2(\eta, T)} \left[ \mathbb{E}_{S \sim \mathcal{D}^n} \Delta_S^{\text{oracle}} + C_x^2 \mu + 2\eta(T+1)\frac{\beta_2}{\mu} \right].$$

*Proof of Theorem 3.6.* Define $t_0 := [\frac{9T}{10}]$ and $k := \frac{1}{\left(1 + \frac{1}{T}\right)\left(1 + \frac{2HC_x^2\eta}{\sqrt{m}}\right)}$. Dividing both sides of

equation (14) by $\left(1 + \frac{1}{T}\right)^{t+1} \left(1 + \frac{2HC_x^2\eta}{\sqrt{m}}\right)^{t+1}$ and summing up, we have

$$2\eta\left(1 + \frac{1}{T}\right) \sum_{t=1}^{T} k^t \left(\widehat{L}_{rob}(\mathbf{W}; S) + 2\eta(T+1)\frac{\beta_2}{\mu}\right) + \|\mathbf{W} - \mathbf{W}_0\|_F^2$$

$$\geq 2\eta\left(1 + \frac{1}{T}\right) \sum_{t=t_0}^{T} k^t M^\mu(\mathbf{W}_t; S).$$

Taking minimum over all weight matrices W,

$$2\eta\left(1 + \frac{1}{T}\right) \sum_{t=t_0}^{T} k^t \left(\widehat{L}_{rob}(\mathbf{W}_t; S) - C_x^2\mu\right)$$

$$\leq 2\eta\left(1 + \frac{1}{T}\right) \sum_{t=t_0}^{T} k^t M^\mu(\mathbf{W}_t; S) \qquad \text{(Lemma C.1-6.)}$$

$$\leq \min_{\mathbf{W}} \left(2\eta\left(1 + \frac{1}{T}\right) \sum_{t=1}^{T} k^t \left(\widehat{L}_{rob}(\mathbf{W}; S) + 2\eta(T+1)\frac{\beta_2}{\mu}\right) + \|\mathbf{W} - \mathbf{W}_0\|_F^2\right)$$

$$\leq 2\eta\left(1 + \frac{1}{T}\right) \sum_{t=1}^{T} k^t \cdot \min_{\mathbf{W}} \left(\widehat{L}_{rob}(\mathbf{W}; S) + \frac{2}{\eta T}\|\mathbf{W} - \mathbf{W}_0\|_F^2 + 2\eta(T+1)\frac{\beta_2}{\mu}\right)$$

$$= 2\eta\left(1 + \frac{1}{T}\right) \sum_{t=1}^{T} k^t \cdot \left(\Delta_S^{\text{oracle}} + 2\eta(T+1)\frac{\beta_2}{\mu}\right).$$

Taking the expectation on both sides, we get

$$\sum_{t=t_0}^{T} k^t \left(\min_{t_0 \leq t \leq T} \mathbb{E}_{S \sim \mathcal{D}^n} \widehat{L}_{rob}(\mathbf{W}_t; S) - C_x^2\mu\right) \leq \sum_{t=t_0}^{T} k^t \left(\mathbb{E}_{S \sim \mathcal{D}^n} \widehat{L}_{rob}(\mathbf{W}_t; S) - C_x^2\mu\right)$$

$$\leq \sum_{t=1}^{T} k^t \cdot \left(\mathbb{E}_{S \sim \mathcal{D}^n} \Delta_S^{\text{oracle}} + 2\eta(T+1)\frac{\beta_2}{\mu}\right).$$

Simplifying the equation above, we get

$$\min_{t_0 \leq t \leq T} \mathbb{E}_{S \sim \mathcal{D}^n} \widehat{L}_{rob}(\mathbf{W}_t; S) \leq C_x^2\mu + \frac{\sum_{t=1}^{T} k^t}{\sum_{t=t_0}^{T} k^t} \cdot \left(\mathbb{E}_{S \sim \mathcal{D}^n} \Delta_S^{\text{oracle}} + 2\eta(T+1)\frac{\beta_2}{\mu}\right) \qquad (15)$$

$$\leq C_x^2\mu + \left(\sum_{r=0}^{9} \left(\frac{1}{k}\right)^{r\frac{T}{10}}\right) \cdot \left(\mathbb{E}_{S \sim \mathcal{D}^n} \Delta_S^{\text{oracle}} + 2\eta(T+1)\frac{\beta_2}{\mu}\right)$$

$$\leq C_x^2 \mu + \left( \sum_{r=0}^{9} e^{\frac{3r}{10}} \right) \cdot \left( \mathbb{E}_{S \sim \mathcal{D}^n} \Delta_S^{\text{oracle}} + 2\eta(T+1)\frac{\beta_2}{\mu} \right)$$

$$(m \geq H^2 C_x^4 \eta^2 T^2)$$

$$\leq C_x^2 \mu + 55 \left( \mathbb{E}_{S \sim \mathcal{D}^n} \Delta_S^{\text{oracle}} + 2\eta(T+1)\frac{\beta_2}{\mu} \right).$$

Equation (4) gives us for any $t \leq T$, $\mathbb{E}_{S \sim \mathcal{D}^n} \varepsilon_{gen}(\mathbf{W}_t) \leq \frac{\alpha_2(\eta, T)}{1-\alpha_2(\eta, T)} \mathbb{E}_{S \sim \mathcal{D}^n} \widehat{L}_{rob}(\mathbf{W}_t; S)$. Therefore,

$$\min_{t_0 \leq t \leq T} \mathbb{E}_{S \sim \mathcal{D}^n} \varepsilon_{gen}(\mathbf{W}_t) \leq \frac{\alpha_2(\eta, T)}{1-\alpha_2(\eta, T)} \min_{t_0 \leq t \leq T} \mathbb{E}_{S \sim \mathcal{D}^n} \widehat{L}_{rob}(\mathbf{W}_t; S)$$

$$\leq \frac{\alpha_2(\eta, T)}{1-\alpha_2(\eta, T)} \left( C_x^2 \mu + 55 \left( \mathbb{E}_{S \sim \mathcal{D}^n} \Delta_S^{\text{oracle}} + 2\eta(T+1)\frac{\beta_2}{\mu} \right) \right).$$

The proof of the second statement takes a similar approach. Following the same procedure, we can replace $t_0$ by 0 in equation (15), and get

$$\min_{0 \leq t \leq T} \mathbb{E}_{S \sim \mathcal{D}^n} \widehat{L}_{rob}(\mathbf{W}_t; S) \leq \mathbb{E}_{S \sim \mathcal{D}^n} \Delta_S^{\text{oracle}} + C_x^2 \mu + 2\eta(T+1)\frac{\beta_2}{\mu}.$$

Combining with equation (4),

$$\min_{0 \leq t \leq T} \mathbb{E}_{S \sim \mathcal{D}^n} L_{rob}(\mathbf{W}_t) \leq \frac{1}{1-\alpha_2(\eta, T)} \min_{0 \leq t \leq T} \mathbb{E}_{S \sim \mathcal{D}^n} \widehat{L}_{rob}(\mathbf{W}_t; S)$$

$$\leq \frac{1}{1-\alpha_2(\eta, T)} \left( \mathbb{E}_{S \sim \mathcal{D}^n} \Delta_S^{\text{oracle}} + C_x^2 \mu + 2\eta(T+1)\frac{\beta_2}{\mu} \right).$$

$\square$

**Corollary 3.7.** After $T \leq \mathcal{O}(\min\{n^2, \frac{1}{\beta_2^{2/3}}\})$ iterations of Algorithm 1 with Moreau Envelope with step-size $\eta = \mu = \Theta(\frac{1}{C_x^2 \sqrt{T}})$ on a network of width $m \geq \Omega(T)$, we have for any weight matrix W,

$$\min_{1 \leq t \leq T} \mathbb{E}_{S \sim \mathcal{D}^n} L_{rob}(\mathbf{W}_t) \leq 1.1 L_{rob}(\mathbf{W}) + \mathcal{O}\left( \frac{C_x^2 \|\mathbf{W} - \mathbf{W}_0\|_F^2}{\sqrt{T}} \right) + \mathcal{O}\left( \frac{1}{\sqrt{T}} \right).$$

*Proof of Corollary 3.7.* Under the conditions of the corollary, we have $m \geq H^2 C_x^4 \eta^2 T^2$, and $\alpha_2(\eta, T) = \mathcal{O}(C_x^2 \frac{\eta T}{n} + C_x \eta T \sqrt{\frac{\beta_2}{\mu}})$ can be small enough so that $\frac{1}{1-\alpha_2(\eta, T)} \leq 1.1$. Then it is clear that this corollary is a special case of Theorem 3.6. $\square$

