# OpenReview forum: "Stability and Generalization of Adversarial Training for Shallow Neural Networks with Smooth Activation"
_NeurIPS.cc/2024/Conference — NeurIPS 2024 poster_

### Official Review · Reviewer_RLPb · 2024-06-26

**Soundness:** 2
**Presentation:** 3
**Contribution:** 3
**Rating:** 7
**Confidence:** 3

**Summary:**

The paper studies generalization bounds for two-layer networks trained using several variants of adversarial training. The main results are a bound on the stability of the algorithm, which in turn provides a bound on the generalization and the robust accuracy.

**Strengths:**

- As far as I know, the results presented in this paper are novel. The approach of studying the generalization capabilities of adversarial training through the lens of stability seems like a promising direction with interesting results.

- The paper studies 2-layer networks (where the weights of the second layer are fixed) which go beyond current research in this field, and potentially beyond the NTK/kernel regime.

- The results of the Moreau Envelope are also novel in this context AFAIK, and provide tighter bounds than GD or SGD analysis (although they are computationally less efficient)

**Weaknesses:**

Although the results of this paper are interesting, there are some issues with the proofs and with some of the setting of the problem, specifically the amount of over-parameterization. In more detail:

1) The assumption on m I think is a bit misleading. If I understand correctly, the authors assume that $m \geq T^2 \eta^2$ which means that the change of the weights when doing GD is very small compared to the number of neurons. This in turn means that slightly changing the loss (by a single sample), and beginning from the same starting point will lead to almost the same weights, hence the stability result. However, I don’t think this assumption makes sense in more practical settings. It basically means that the number of iterations is so small (or the learning rate is so small) that most weights barely move from their initialization.
This is also emphasized in Theorem B.1, where the change depends on $\exp(T\eta /\sqrt(m))$. This means that even if m is slightly smaller than $T^2\eta^2$, (e.g. $m = (T\eta)^{1.5}$) then the difference changes exponentially, and the algorithm is very non-stable. This issue is not addressed in the paper and is only briefly mentioned in the proof intuition.

2) The bounds in Theorem 3.2 don’t make sense. If $\alpha_1(\eta,T) > 1$ then the r.h.s of the bounds is negative. Do the bounds only work when $\alpha_1(\eta,T) < 1$? If so, this means that $T<1/\eta\cdot\beta_1$ so the number of iterations might be very small compared to the learning rate.
This also seems relevant to Theorem 3.6 with $\alpha_2$

3) There is an issue with the value of T in Corollary 3.3. Which value of T gives that $1/(1/\alpha_1) < 1.1$? Suppose $\beta_1$ is a constant, then we need $T<1/\eta$ to get a constant term rather than $T< 1/\eta^2$, no?

4) Line 509 - I don’t understand this, how is Eq (4) related to \epsilon_gen?

I will be happy to see the author’s response to these issues and will consider raising my score accordingly.

**Questions:**

1) Given $\epsilon > 0$, how should we set the parameters of the problem (i.e. $T, \beta_1, \eta, ||W-W_0||, m$ etc.) to get generalization error and robust accuracy smaller than $\epsilon$?

2) What happens when m is chosen independently of T - although we allow it to depend on n, i.e. the overparameterized regime compared to the number of samples.

**Limitations:**

The authors discuss the limitations of their results.

---

> ### Author Rebuttal · Authors · 2024-08-05
>
> ### W1: Regarding $m\geq  \eta^2 T^2$
>
> Reviewer has a correct understanding of why stability holds, i.e., the change of the weights when doing GD is very small compared to the number of neurons. We utilize the weakly convex robust loss (see Lemma 4.1) as well as $m\geq O(\eta^2 T^2)$ to establish stability.(see Theoorem 4.3)
>
> Note that training neural networks (i.e., implementing the ERM rule) is known to be computationally hard, even for a two layer network with only three hidden nodes. This has been known since the early 90s. Recent results in deep learning theory focus on two-layer neural networks that are greatly overparameterized ($m$ is large) and involve a certain scale of initialization. Under these settings, we can bound the training error of GD after T iterations by $1/\sqrt{T}$. This is the state-of-the-art results in computational theory of deep learning.
>
> Now, the important point we want to make here is that the key to the analysis is ensuring that the weights of the trained network do not change by much from initialization. One can then show that training neural networks is similar to dynamics of kernel methods. This is the dominant framework for computational learning theoretical guarantees for training neural networks and is aptly called the “lazy regime” or the “neural tangent kernel” (NTK) setting.
>
> The early works that laid the foundations for this theory are [r1, r2, r3].
>
> ### W2: Regarding $\alpha_1(\eta,T)<1$
>
> Yes, for that result of ours to make sense we do need $\alpha_1(\eta,T) < 1$. However, we can also give a result that is of the same form as the expression in equation (4) of [r4] and get a factor of  $C\alpha_1(\eta,T)\cdot (1+\alpha_1(\eta,T))$ instead of $C\frac{\alpha_1(\eta,T)}{1-\alpha_1(\eta,T)}$. In that case we don’t need $\alpha_1(\eta,T) < 1$.
>
> ### W3: Regarding $\frac{1}{1-\alpha_1}<1.1$
>
> It is possible that the reviewer did not realize $\eta=O(\frac{1}{\sqrt{T}})$.
>
> In order to get $\frac{1}{1-\alpha_1}<1.1$, we need $\alpha_1$ to be a small positive real number, and equivalently, $\eta\sqrt{T}$, $\frac{\eta T}{n}$ and $\sqrt{\beta_1\eta T}$ are small positive real numbers (through the definition of $\alpha_1$). Note that we have selected $\eta=c_0\frac{1}{\sqrt{T}}$ in Corollary 3.3, plugging in $\eta$, we equivalently require $c_0$, $c_0\frac{\sqrt{T}}{n}$ and $c_0\beta_1\sqrt{T}$ to be small positive real numbers. In Corollary 3.3, we assumed $T\leq O(n^2)$ and $T\leq O(1/\beta_1^2)$, so we can choose $c_0$ to be a proper (small) positive constant so that the three inequalities hold.
>
> ### W4: Regarding line 509
>
> The definition of the generalization gap is the test loss minus the training loss $\epsilon_{gen}=L_{rob}(W_t)-$$\hat{L}_{rob}(W_t;S)$
>
> so we can get line 509 by subtracting $E_{S\sim\mathcal{D}^{n}}\hat{L}_{rob}(W_t;S)$ on both sides of Eq (4), so the left hand side will have the generalization gap.
>
> ### Q1: Given $\epsilon>0$, how should we set the parameters of the problem to get generalization error and robust accuracy smaller than $\epsilon$?
>
> The relationship is a bit involved since our bounds are post-hoc and not a-priori. In other words, our bound is instance specific, and depends on the given training data and the output of the algorithm. This is in stark contrast to bounds that are based on uniform convergence and hold simultaneously for all hypotheses in the class – there you can ask how many samples or iterations you need to ensure suboptimality. However, note that uniform convergence bounds can be overly pessimistic/vacuous.
> Anyhow, even though there is not a simple answer, below we try to give an intuitive understanding.
>
> Let’s say you want to bound the expected robust test loss by $(1+\epsilon)$ times the expected training loss (see Lemma 4.2 relating the two quantities). Then,
> To ensure $\frac{1}{1-C_x\cdot \alpha_1}\leq 1+\epsilon$, we need $\alpha_1\leq\frac{\epsilon}{C_x(1+\epsilon)}=O(\epsilon)$. We can give two possible ways of setting the different parameters to ensure that $\alpha_1 = O(\epsilon)$.
>
> Recall that $\alpha_1=O(\eta\sqrt{T}+\frac{\eta T}{n}+\sqrt{\beta_1\eta T})$.
>
> Set $\beta_1 = O(\epsilon^2)$, $n=\Theta(1/\epsilon)$, $T=\Theta(1/\epsilon^2)$, $\eta=O(\frac{1}{T})$,
>
> or set $\beta_1 = O(\epsilon^3)$, $n=\Theta(1/\epsilon^2)$, $T=\Theta(1/\epsilon^4)$, $\eta=O(\frac{\epsilon}{\sqrt{T}})$.
>
> Check that $\eta\sqrt{T}=O(\epsilon)$, $\frac{\eta T}{n}=O(\epsilon)$ and $\sqrt{\beta_1\eta T}=O(\epsilon)$.
>
> ### Q2: What happens when m is chosen independently of T
>
> We can choose $m \geq \Theta(n^2)$, and then set $\eta T\leq O(n)$. Note though that it seems superfluous as there could be other regimes that could still give us the condition we need. This is why prior works (e.g., [r4, r5]) also state the assumption as we do.
>
> This also makes sense as our generalization bounds are based on algorithmic stability which depends on parameters $\eta$ and $T$. If we were using tools from uniform convergence, then m would naturally depend on the sample size and input dimension d.
>
> [r1] Du, Simon S., et al. "Gradient descent provably optimizes over-parameterized neural networks." arXiv preprint arXiv:1810.02054 (2018).
>
> [r2] Arora, Sanjeev, et al. "Fine-grained analysis of optimization and generalization for overparameterized two-layer neural networks." International Conference on Machine Learning. PMLR, 2019.
>
> [r3] Ji, Ziwei, et al. "Polylogarithmic width suffices for gradient descent to achieve arbitrarily small test error with shallow relu networks." arXiv preprint arXiv:1909.12292 (2019).
>
> [r4] Richards, Dominic, et al. "Stability & generalisation of gradient descent for shallow neural networks without the neural tangent kernel." Advances in neural information processing systems 34 (2021): 8609-8621.
>
> [r5] Lei, Yunwen, et al. "Stability and generalization analysis of gradient methods for shallow neural networks." Advances in Neural Information Processing Systems 35 (2022): 38557-38570.

---

> ### Comment · Reviewer_RLPb · 2024-08-12
>
> I thank the authors for their response.
>
> Other reviewers (including myself) found it strange that the overparameterization depends on the learning rate and number of iterations. However, after reading the author's response I agree that this dependence is indeed the right one for stability analysis of generalization bounds, contrary to uniform convergence analysis.
>
> To conclude, my questions are answered, and I will raise my score to 7. I think the authors should add in the final version several clarifications that were raised from the reviewer's question. I also think it would strengthen the paper to include the proof for \alpha_1  > 1.

---

### Official Review · Reviewer_381R · 2024-07-12

**Soundness:** 3
**Presentation:** 3
**Contribution:** 3
**Rating:** 6
**Confidence:** 3

**Summary:**

Adversarial Training is a popular method to train models that enhance robutness to adversarial examples. In recent years, a lot of papers study the generalization of adversarial training in various models. In this paper, the authors study the generalization of adversarial training for a special shallow neural networks with uniform stability, they show the theoretical reults for three different algorithms.

**Strengths:**

1. The authors show theoretical results for a special shallow neural networks with logistic loss, which is non-convex and non-smooth.
2. The authors discuss stability and generalization guarantees of three variants of adversarial training.
3. The proof of this work is clear to read.

**Weaknesses:**

1. The model studied in this work is a special neural network. By fixing the weights of last layer, it has only one layer of trainable parameters, it is unclear if the results shown in this special neural network can be generalized to regular neural netowrk.
2. The authors claim that they use a over-parameterized neural network, while they explian "over-parameterized" as $m \geq O(\eta T)$. It is confused that $m$ depends on $\eta$ and $T$ but not depends on the number and dimention of inputs.
3. In section 4, the bound shown in theoretical results depend on $m$ with a same order. I check the proof of this work, and find that these bounds depend on $m$ since the weights of last layer is initilized from $\\{ \frac{1}{\sqrt{m}}, -\frac{1}{\sqrt{m}} \\}$, which means that the bounds depends on the initilization of weights of last layer. If the last layer is initilized from other parameters, especially, initilized from $\\{ \sqrt{m}, -\sqrt{m} \\}$, does the reult still hold for over-parameterized neural network?
4. The main techniques used in proofs depend on the weakly convex property of function, and uniform stability of algorithm. It is suggested to show the formal definition of weakly convex function.

**Questions:**

Is this possible to generalize the results from special neural network to regular neural network with mild assumptions like the Theorem 4.7 in [1]?

**Limitations:**

This work study a speical neural network, and it is unclear whether the results can be generalized to regular neural netowrks.

---

> ### Author Rebuttal · Authors · 2024-08-05
>
> ### W1: The model studied in this work is a special neural network. By fixing the weights of last layer, it has only one layer of trainable parameters, it is unclear if the results shown in this special neural network can be generalized to regular neural netowrk.
>
> It is known since the early 90s that training neural networks is computationally hard, even for two layer networks with three hidden neurons. Modern theory of deep learning avoids those hardness results by considering (a) overparameterized two layer-networks, (b) a certain small-scale randomized initialization, and (c) freezing the top layer and training only the bottom layer. Under this setting, we can bound the training error of GD after T iterations by $1/\sqrt{T}$. This is the state-of-the-art results in computational theory of deep learning. Yes, it may be too specialized and not close to practice, but this is all we have currently. So, even in the standard setting, it remains unclear if these results can be generalized to “regular” networks.
>
>
> ### W2: The authors claim that they use a over-parameterized neural network, while they explian "over-parameterized" as $m\geq O(\eta T)$. It is confused that $m$ depends on $\eta$ and $T$ but not depends on the number and dimention of inputs.
>
> We state $m$ in terms of $T$ and the learning rate \eta as our generalization bounds are based on algorithmic stability which depends on those algorithmic parameters. Similar assumptions have also appeared in related prior works [r1] and [r2].
> If we were using tools from uniform convergence, then m would naturally depend on the sample size and input dimension $d$. Note that we could still write the condition $m \geq O(\eta^2 T^2)$ as $m \geq \Theta(n^2)$ and $\eta T = O(n)$, but that would seem superfluous as there could be other regimes that could still give us the condition we need.
>
> [r1] Richards, Dominic, and Ilja Kuzborskij. "Stability & generalisation of gradient descent for shallow neural networks without the neural tangent kernel." Advances in neural information processing systems 34 (2021): 8609-8621.
>
> [r2] Lei, Yunwen, Rong Jin, and Yiming Ying. "Stability and generalization analysis of gradient methods for shallow neural networks." Advances in Neural Information Processing Systems 35 (2022): 38557-38570.
>
> ### W3: In section 4, the bound shown in theoretical results depend on $m$ with a same order. I check the proof of this work, and find that these bounds depend on $m$ since the weights of last layer is initilized from $\\{\frac{1}{\sqrt{m}},-\frac{1}{\sqrt{m}}\\}$, which means that the bounds depends on the initilization of weights of last layer. If the last layer is initilized from other parameters, especially, initilized from $\\{{\sqrt{m}},-{\sqrt{m}}\\}$, does the reult still hold for over-parameterized neural network?
>
> No. The scale of the initialization is important. As we state above, this is the case with the dominant framework for a computational theory of deep learning even in the standard setting. Such initialization has also appeared in prior works, e.g., [r3, r4, r5]. In particular, we need the initialization to be at that scale for our Lemma 4.1 to hold.
>
> [r3] Du, Simon S., et al. "Gradient descent provably optimizes over-parameterized neural networks." arXiv preprint arXiv:1810.02054 (2018).
>
> [r4] Arora, Sanjeev, et al. "Fine-grained analysis of optimization and generalization for overparameterized two-layer neural networks." International Conference on Machine Learning. PMLR, 2019.
>
> [r5] Ji, Ziwei, and Matus Telgarsky. "Polylogarithmic width suffices for gradient descent to achieve arbitrarily small test error with shallow relu networks." arXiv preprint arXiv:1909.12292 (2019).
>
>
> ### W4: The main techniques used in proofs depend on the weakly convex property of function, and uniform stability of algorithm. It is suggested to show the formal definition of weakly convex function.
>
> We will include a formal definition of weakly convex functions.
>
> ### Q1: Is this possible to generalize the results from special neural network to regular neural network with mild assumptions like the Theorem 4.7 in [1]?
>
> We are not sure what is reference [1], but see our response above regarding the state-of-affairs with computational learning theoretic results for training neural networks.

---

> > ### Comment · Reviewer_381R · 2024-08-12
> >
> > Thanks for your response. My main concern is regarding the setting of an over-parameterized neural network. After reading the authors' response and the comments from other reviewers, my concerns have been addressed. I will raise my rating to a "Weak Accept" accordingly.

---

### Official Review · Reviewer_GasY · 2024-07-13

**Soundness:** 4
**Presentation:** 3
**Contribution:** 2
**Rating:** 6
**Confidence:** 4

**Summary:**

This work studies the stability of adversarial training in two-layer neural networks in binary classification problems. The authors study gradient-descent-based adversarial training, with nearly-optimal adversarial perturbations, in two-layer neural networks with a frozen second layer, and they obtain guarantees on the stability of the algorithm and its robust generalization. Furthermore, the paper considers a smoothened empirical robust loss and obtains guarantees for this case too. The proofs of the main results are deferred to the Appendix, yet a proof sketch is presented in Section 4 for the case of (exactly) optimal adversarial perturbations during training.

**Strengths:**

The paper is well written and makes it easy to follow its contributions and contextualize them with respect to prior work. Furthermore, the results are cleanly organized and presented. The main result is a stability bound on GD-adversarial training, which overcomes the non-convexity and non-smoothness of the objective. I particularly enjoyed reading the proof sketch section and, in particular, Lemma 4.1, which makes it clear how the analysis can sidestep the aforementioned challenges. I consider this work to be a valuable contribution to the field of theoretical robust (deep) learning.

**Weaknesses:**

The theoretical results hold for a special class of networks (two-layer with smooth activations and frozen 2nd layer weights) and do not appear to be particularly surprising or useful for inspiring practical ideas. This is the reason why I do not recommend a higher score. Perhaps the authors could discuss some of their assumptions in Section 5; for instance, they could comment on the assumption of eq. (1). Should we consider $\beta_1$ to be constant during training? Do we expect it to be smaller or larger as training progresses? Could it be empirically estimated? Similarly, the authors could comment on the feasibility of adversarial training with the smoothened loss.

**Questions:**

I have a few questions/suggestions:

* Lines 9 and 10 in the abstract: If I understand correctly, there should be something added along the lines of “provided there exists a robust network around the initialization.”
* Lines 193-194, “This view is also consistent with several empirical studies.”: can you please elaborate on this?
* Theorem 3.2, for the result on the generalization gap you should explicitly state the loss function, right? (since eq. (3) is defined with respect to a generic function $f$).

**Limitations:**

The limitations of the work are thoroughly discussed.

---

> ### Author Rebuttal · Authors · 2024-08-05
>
> ### W1: Detached from practice?
>
> Our is a theoretical result and theory always lags practice and rarely matches the practice perfectly. Having said that, we would like to remind the reviewer that it is known since the early 90s that training neural networks is computationally hard, even for two layer networks with three hidden neurons. Modern theory of deep learning avoids those hardness results by considering (a) overparameterized two layer-networks, (b) a certain small-scale randomized initialization, and (c) freezing the top layer and training only the bottom layer. Under this setting, we can bound the training error of GD after T iterations by $1/\sqrt{T}$. This is the state-of-the-art results in computational theory of deep learning. It is indeed too specialized and not close to practice, but this is all we have currently.
>
> ### W2: Feasibility of adversarial training with the smoothed loss.
>
> Yes, it is very feasible and practical and has shown to be successful in prior work. [Xiao 2024] discuss at length the feasibility of adversarial training with smoothed loss and present extensive empirical results.
>
> [Xiao 2024] "Uniformly Stable Algorithms for Adversarial Training and Beyond." arXiv preprint arXiv:2405.01817 (2024).
>
> ### W3: How to think about $\beta_1$?
>
> We should not think of $\beta_1$ as a constant. It is a parameter that you as an algorithm designer can choose. You can make it arbitrarily small by adding more computation. Large $\beta_1$ means that the quality of simulated adversarial attacks is poor, so indeed our bounds suggest that the robust generalization will not be good. This is what should be expected, it is not a weakness of our result.
> We further expand on our comment above. Recall that adversarial training involves finding an adversarial perturbation of every training example in the training set. Can we solve this optimization problem exactly? If so, then $\beta_1$ is equal to zero. Most works in theory of robust learning assume that. We argue that in practice that is not true. So, we allow for approximate optimization for finding an adversarial example during training. We introduce a parameter $\beta_1$ that is bound on the suboptimality of an adversarial example. Note that this is a design parameter that a practitioner can choose. If you choose a large beta, the training is easier but the resulting model is not good. So, of course, one should choose a small beta. This involves a computational cost that most previous papers ignore. The only paper that carefully shows how many iterations are needed to find a $\beta_1$ suboptimal attack is that of Mianjy and Arora (2023) – they show that we need $1/\beta_1^2$ iterations of PGD attack to find a $\beta_1$ suboptimal attack.
>
> [Mianjy and Arora 2023] “Robustness Guarantees for Adversarially Trained Neural Networks,” NeurIPS 2023.
>
> ### Q1: Lines 9 and 10 in the abstract: If I understand correctly, there should be something added along the lines of “provided there exists a robust network around the initialization.”
>
> While we give a bound on robust generalization in terms of distance from initialization, it does not mean that it is a necessary and the only condition for robust generalization. In fact, we can also give a robust generalization bound that does not depend on the distance from initialization. This follows using Theorem 3.1 and Lemma 4.2.
>
> ### Q2: Lines 193-194, “This view is also consistent with several empirical studies.”: can you please elaborate on this?
>
> Early stopping is a standard approach for algorithmic regularization. It has been shown to prevent overfitting in many settings, see for example [r1, r2, r3]. Here, we show that it is able to mitigate robust overfitting as well.
>
> [r1] ​​Caruana, Rich, Steve Lawrence, and C. Giles. "Overfitting in neural nets: Backpropagation, conjugate gradient, and early stopping." Advances in neural information processing systems 13 (2000).
>
> [r2] Rice, Leslie, Eric Wong, and Zico Kolter. "Overfitting in adversarially robust deep learning." International conference on machine learning. PMLR, 2020.
>
> [r3] Pang, Tianyu, et al. "Bag of tricks for adversarial training." arXiv preprint arXiv:2010.00467 (2020).
>
> ### Q3: Theorem 3.2, for the result on the generalization gap you should explicitly state the loss function, right? (since eq. (3) is defined with respect to a generic function $f$).
>
> Yes. We will state the loss function explicitly.

---

> > ### Comment · Reviewer_GasY · 2024-08-13
> >
> > I apologise for my delayed response.
> >
> > Thank you very much for your reply and the references provided (especially [Mianjy and Arora 2023]). I would advise you to reference this paper when introducing $\beta_1$, and to discuss its computational considerations.

---

### Official Review · Reviewer_fRZq · 2024-07-13

**Soundness:** 3
**Presentation:** 3
**Contribution:** 3
**Rating:** 6
**Confidence:** 3

**Summary:**

This paper uses uniform stability to analyze adversarial training on wide shallow networks when the adversarial perturbations are $\beta_1$-optimal. Assuming there exists a robust network near initialization, in expectation the best network iterate has test loss that scales with $1/\sqrt{T}$. The results for GD are extended to SGD and when using Moreau's envelope.

**Strengths:**

- The bounds are tied to how strong the adversarial training attack is, providing an approximation to practical attacks while allowing for tractability in the theoretical analysis.
- No assumptions are made on the data distribution, and the adversarial attack threat model is very general.

**Weaknesses:**

- The bounds are in expectation and not high probability bounds.
- A large width is required for the bounds to hold.
- The $\beta_1$ parameter is set in the worst-case, and as a result for practical attack algorithms is likely to be high. In fact, if $\beta_1 > 0$ is constant, then it appears Corollary 3.3 is vacuous.

**Questions:**

- How realistic is the assumption that there exists a robust network in the vicinity of the initialization?
- Should I be thinking of $beta_1$ as a constant? If so, how should Corollary 3.3 be interpreted?

**Limitations:**

Yes.

---

> ### Author Rebuttal · Authors · 2024-08-05
>
> ### W1: The bounds are in expectation and not high probability bounds.
>
> We can give high probability bounds based on [r1], which relates high probability generalization bounds with algorithmic stability, but they are looser than the bounds in expectation. We had those originally in the paper but chose to remove them as they are not very informative.
>
>
>
> ### W2: A large width is required for the bounds to hold.
>
> Training neural networks (i.e., implementing the ERM rule) is known to be computationally hard, even for a two layer network with only three hidden nodes. This has been known since the 90s. Recent results in deep learning theory focus on the setting where networks are greatly overparameterized. Indeed, all results giving computational guarantees for learning deep neural networks assume that the networks are overparameterized. So, it is not surprising that we need a similar condition in the adversarially robust setting. The assumption on width, $m\geq \eta^2 T^2$, has also appeared in related prior works [r2] and [r3].
>
> Also, note that the way we present the results, we can also rewrite the condition above as $\eta T \leq O(\sqrt{m})$ and interpret it as early stopping .
>
> ### W3: The $\beta_1$ parameter is set in the worst-case, and as a result for practical attack algorithms is likely to be high. In fact, if $\beta_1>0$ is constant, then it appears Corollary 3.3 is vacuous.
>
> [TLDR] $\beta_1$ is not a constant, it is a parameter you can choose. You can make it arbitrarily small by adding more computation. Large $\beta_1$ means that the quality of simulated adversarial attacks is poor, so indeed our bounds suggest that the robust generalization will not be good. This is what should be expected, it is not a weakness of our result.
>
> Recall that adversarial training involves finding an adversarial perturbation of every training example in the training set. Can we solve this optimization problem exactly? If so, then $\beta_1$ is equal to zero. Most works in theory of robust learning assume that. We argue that in practice that is not true. So, we allow for approximate optimization for finding an adversarial example during training. We introduce a parameter $\beta_1$ that is bound on the suboptimality of an adversarial example. Note that this is a design parameter that a practitioner can choose. If you choose a large $\beta_1$, the training is easier but the resulting model is not good. So, of course, one should choose a small $\beta_1$. This involves a computational cost that most previous papers ignore. The only paper that carefully shows how many iterations are needed to find a $\beta_1$ suboptimal attack is that of Mianjy and Arora (2023) – they show that we need $1/\beta_1^2$ iterations of PGD attack to find a $\beta_1$ suboptimal attack.
>
> [Mianjy and Arora, 2023] “Robustness Guarantees for Adversarially Trained Neural Networks,” NeurIPS 2023.
>
> ### Q1: How realistic is the assumption that there exists a robust network in the vicinity of the initialization?
>
> [TLDR] Existence of a network that generalizes well in the vicinity of a certain way of initialization is a high dimensional phenomenon that is central to the NTK setting, a dominant framework for computational learning theory of deep neural networks.  Nonetheless, our results also yield guarantees that do not depend on the distance from initialization.
>
> Most computational learning results for deep neural networks assume overparametrization and a certain initialization that ensures that the weights of the trained network do not change by much from initialization. Yet, these trained networks in the so-called lazy regime or the NTK setting, are guaranteed to generalize. So, assuming the existence of a network that generalizes well in the vicinity of initialization (chosen at the right scale) is a high dimensional phenomenon. We argue that if the conditions of our theorems are met (i.e., we consider over-parametrized settings), then such an assumption is not unrealistic. It is same as saying “how realistic is it that a unit ball has almost no volume and all of its mass is concentrated near the boundary.” Well it happens in high dimensions.
>
> While we give a bound on robust generalization in terms of distance from initialization, it does not mean that it is a necessary and the only condition for robust generalization. In fact, we can also give a robust generalization bound that does not depend on the distance from initialization. This follows using Theorem 3.1 and Lemma 4.2.
>
> ### Q2: Should I be thinking of $\beta_1$ as a constant? If so, how should Corollary 3.3 be interpreted?
>
> No, $\beta_1$ should not be treated as constant, it is a design parameter you can choose. Please see a detailed discussion above.
>
> [r1] Feldman, Vitaly, and Jan Vondrak. "High probability generalization bounds for uniformly stable algorithms with nearly optimal rate." Conference on Learning Theory. PMLR, 2019.
>
> [r2] Richards, Dominic, and Ilja Kuzborskij. "Stability & generalisation of gradient descent for shallow neural networks without the neural tangent kernel." Advances in neural information processing systems 34 (2021): 8609-8621.
>
> [r3] Lei, Yunwen, Rong Jin, and Yiming Ying. "Stability and generalization analysis of gradient methods for shallow neural networks." Advances in Neural Information Processing Systems 35 (2022): 38557-38570.

---

> > ### Comment · Reviewer_fRZq · 2024-08-14
> >
> > Thank you for addressing my concerns, especially regarding $\beta_1$. I have raised my score to a 6.
> >
> > I wonder if requiring the $\beta_1$-optimal condition to hold everywhere can be relaxed (for example, to an averaged case), for networks where guaranteeing nearly optimal adversarial attacks is more difficult.

---

### Decision · Program_Chairs · 2024-09-25

**Decision:**

Accept (poster)

**Comment:**

This paper studies the stability and generalization of adversarial training for two-layer neural networks with smooth activation functions in binary classification problems, focusing on settings beyond the NTK regime and without data distribution assumptions. The authors derive generalization bounds based on uniform stability, particularly for gradient descent (GD) and stochastic gradient descent (SGD). After the authors’ responses, the reviewers recognized the theoretical novelty and generally agreed that this work contributes to a valuable theoretical understanding of adversarial training. Some reviewers also raised concerns about the practical relevance of the results, though this concern is common in theoretical papers. Overall, all reviewers appreciate this theoretical paper, and I recommend acceptance.